# Learning single-cell perturbation responses using neural optimal transport

Charlotte Bunne [1,2,9], Stefan G. Stark[1,2,3,4,9], Gabriele Gut [5,9], Jacobo Sarabia del Castillo[5], Mitch Levesque[6], Kjong-Van Lehmann [1,7] ✉, Lucas Pelkmans [5] ✉, Andreas Krause [1,2] ✉ & Gunnar Rätsch [1,2,3,4,8] ✉

Understanding and predicting molecular responses in single cells upon chemical, genetic or mechanical perturbations is a core question in biology. Obtaining single-cell measurements typically requires the cells to be destroyed. This makes learning heterogeneous perturbation responses challenging as we only observe unpaired distributions of perturbed or non-perturbed cells. Here we leverage the theory of optimal transport and the recent advent of input convex neural architectures to present CellOT, a framework for learning the response of individual cells to a given perturbation by mapping these unpaired distributions. CellOT outperforms current methods at predicting single-cell drug responses, as profiled by scRNA-seq and a multiplexed protein-imaging technology. Further, we illustrate that CellOT generalizes well on unseen settings by (1) predicting the scRNA-seq responses of holdout patients with lupus exposed to interferon-β and patients with glioblastoma to panobinostat; (2) inferring lipopolysaccharide responses across different species; and (3) modeling the hematopoietic developmental trajectories of different subpopulations.

Characterizing and modeling perturbation responses at the single-cell level from non-time-resolved data remains one of biology's grand challenges. It finds applications in predicting cellular reactions to environmental stress or a patient's response to drug treatments. Accurate inference of perturbation responses at the single-cell level allows us to understand how and why individual tumor cells evade cancer therapies[1]. More generally, it deepens the mechanistic understanding of the molecular machinery that determines the respective responses to perturbations. Single-cell responses to genetic or chemical perturbations are highly heterogeneous[2] due to multiple factors, including pre-existing variability in the abundance and subcellular organization of messenger RNA and proteins[3-6], cellular states[7] and the cellular microenvironment[8]. To effectively predict the drug response of each cell in a population, whether derived from tissue culture or as primary cells from a patient biopsy, it is thus crucial to incorporate this heterogeneous multivariate subpopulation structure into the analysis.

A fundamental difficulty in learning perturbation responses is that cells are usually fixed and stained or chemically destroyed to obtain these measurements. Hence, it is only possible to measure the same cells before or after a perturbation is applied. Therefore, while we do not have access to a set of paired control/perturbed single-cell observations, we do have access to separate sets of single-cell observations from control and perturbed cells, respectively. To subsequently match single cells between conditions and, at the same time, account for cellular heterogeneity is a highly complex pairing problem.

Here, we seek to learn a perturbation model that robustly describes the cellular dynamics upon intervention while still accounting for underlying variability across samples. Learning the responses on an

[1]Department of Computer Science, ETH Zurich, Zürich, Switzerland. [2]AI Center, ETH Zurich, Zürich, Switzerland. [3]Medical Informatics Unit, University of Zurich Hospital, Zürich, Switzerland. [4]Swiss Institute of Bioinformatics, Zurich, Switzerland. [5]Department of Molecular Life Sciences, University of Zurich, Zürich, Switzerland. [6]Department of Dermatology, University of Zurich Hospital, University of Zurich, Zürich, Switzerland. [7]Cancer Research Center Cologne–Essen, Site: Center Integrated Oncology Aachen, Aachen, Germany. [8]Department of Biology, ETH Zurich, Zürich, Switzerland. [9]These authors contributed equally: Charlotte Bunne, Stefan G. Stark, Gabriele Gut. ✉e-mail: kjlehmann@ukaachen.de; lucas.pelkmans@mls.uzh.ch; krausea@ethz.ch; gunnar.raetsch@inf.ethz.ch

existing patient cohort enables inference of treatment responses for new (previously unseen) patients, assuming that we captured the heterogeneous drug reactions of patients during training. It is crucial, however, to not simply model average perturbation responses of a patient cohort, but to capture the specificities of a single patient through personalized treatment effect predictions.

Previous methods to approximate single-cell perturbation responses fall short of solving this highly complex pairing problem while, at the same time, accounting for cellular heterogeneity and the strong subpopulation structure of cell samples[9–11]. Current state-of-the-art methods[12–14] predict perturbation responses via linear shifts in a learned latent space. While this can capture nonlinear cell-type-specific responses, the use of linear interpolations reduces the alignment problem to the possibly more challenging task of learning representations that are invariant to the corresponding perturbation.

In this work, we introduce CellOT, a new approach that predicts perturbation responses of single cells by directly learning and uncovering maps between control and perturbed cell states, thus explicitly accounting for heterogeneous subpopulation structures in multiplexed molecular readouts. Assuming perturbations incrementally alter molecular profiles of cells, such as gene expression or signaling activities, we learn these changes and alignments using optimal transportation theory (OT)[15]. Optimal transport provides natural geometric and mathematical tools to manipulate probability distributions. It has found recent successes modeling cellular development processes[16,17], albeit in a non-parameterized setting. Thus, current OT-based approaches are unable to make predictions on unseen cells, such as those from unseen samples, for example from new patients.

Based on recent developments in neural optimal transport[18], CellOT learns an optimal transport map for each perturbation in a fully parameterized and highly scalable manner. Instead of directly learning a transport map[19–21], CellOT parameterizes a pair of dual potentials with input convex neural networks[22]. This choice induces an important theory-motivated inductive bias essential to model stability[18].

We demonstrate CellOT's effectiveness by (1) learning single-cell marker responses to different cancer drugs in melanoma cell lines; (2) predicting single-cell transcriptome responses in biopsies of patients with systemic lupus erythematosus as well as panobinostat treatment outcomes of glioblastoma patients; (3) inferring lipopolysaccharide (LPS) responses across different animal species; and (4) modeling the transcriptome evolution of cell fates in hematopoiesis. Moreover, we benchmark CellOT against current state-of-the-art methods on multiple tasks[12,13].

## Results

### Predicting perturbation responses via optimal transport maps

Small molecule drugs can have profound effects on the cellular phenotype by, for instance, altering signaling cascades. Most of these effects depend on the context in which the perturbation occurs. Given the heterogeneity among single cells in cell populations and tissues, predicting cellular responses requires understanding the rules by which context shapes genome activity and its response to drugs. High-dimensional single-cell data measured via single-cell genomics or multiplexed imaging technologies can provide this contextual information but only return unpaired or unaligned observations of cell populations. Here, CellOT allows us to utilize such unpaired data and enables learning cell-state transitions upon perturbation.

In formal terms, we denote the unperturbed control population by $\rho_c$ consisting of $n$ cells $x_i$ for $i = 1, \ldots, n$. Upon perturbation $k$, the multivariate state of each cell $x_i$ of the unperturbed population changes, which we observe as the perturbed population $\rho_k$ (Fig. 1a). To understand the mode of action and effect of perturbations, we seek to learn the transition and alignment between populations $\rho_c$ and $\rho_k$ via parameterizing a map $T_k$ (see Fig. 1a,b), which explains the transition

of each cell from the unperturbed cell population $\rho_c$ into their perturbed state $\rho_k$ upon treatment $k$. Despite originating from different observations, map $T_k$ determines for each cell $x_i$ the most likely corresponding cell $T_k(x_i)$ in the perturbed population (Fig. 1c). Finding this map then not only allows us to model single-cell trajectories upon perturbation but also to predict the perturbed state of previously unseen control cells. As a result, we can forecast the outcome of a perturbation $k$ by applying the learned map $T_k$ to a new unperturbed population $\rho_c'$ (Fig. 1d).

The optimal map $T_k$ aligning the control and perturbed population, which we seek to find, should best describe the incremental changes in the multivariate profile of each cell after applying a perturbation $k$. Using OT[23,24] to recover these maps and unveil single-cell reprogramming trajectories has been proposed as a strong modeling hypothesis in the domain of single-cell biology[16,17,25–28]. OT problems return the alignment between distributions $\rho_c$ and $\rho_k$ corresponding to the minimal overall cost between aligned molecular profiles, thus determining the most likely state of each cell upon perturbation (Fig. 1c). $T_k$ is learned such that its image corresponds to $\rho_k$ and mass is moved from $\rho_c$ into $\rho_k$ according to a principle of minimal effort. As directly parameterizing the OT map $T_k$[20,21,29] is unstable[18], we parameterize the convex potentials of the dual optimal transport problem $f$ and $g$ by input convex neural networks[22] and recover the optimal map $T_k$ using the gradient of a convex function $g_k$ ($\nabla g_k$)[18]. Supplementary Section A.3 provides a more detailed review of optimal transport methods proposed for single-cell biology problems and how our approach deviates from previous methods.

To put CellOT's performance in perspective, we benchmark it against current state-of-the-art methods based on autoencoders[12,13], which attempt to add perturbation effects through the manipulation of a learned latent representation (reviewed in Supplementary Section A.1). To further test the hypothesis of the OT modeling prior, we compare the learned OT map $\nabla g_k$ for each perturbation $k$ with naive non-OT-based alignments.

### CellOT outperforms state-of-the-art methods

We apply CellOT to predict the responses of cell populations to cancer treatments using a proteomic dataset consisting of two melanoma cell lines (M130219 and M130429)[30], profiled by 4i[5] and a single-cell RNA-sequencing (scRNA-seq) dataset[31], which contain 34 and 9 different treatments, respectively. For more details on the datasets see Online Methods. We benchmarked CellOT against two autoencoder-based tools, scGEN[13] and cAE[12], as well as PopAlign[32], a method based on aligning subpopulations of the control and treated space approximated through a mixture of Gaussian densities. Due to the high-dimensional nature of scRNA-seq data, we apply CellOT on latent representations learned by an autoencoder. The marginal distributions for observed and predicted cell populations for two 4i treatments and two scRNA-seq treatments are shown in Fig. 2a,d. Two features are selected for each perturbation and the complete set of marginals is shown in Supplementary Figs. 1–4. While the autoencoder baselines tend to capture the mean of the treated cell population, they are less successful in matching all heterogeneous states of the perturbed population (higher moments of the perturbed population). Thus, these models tend to learn over-simplified perturbation effects and are insufficient when aiming to understand heterogeneous rather than average cellular behaviors. CellOT, on the other hand, is able to capture these higher moments, yielding accurate and nuanced predictions.

This can be further quantified through distributional metrics such as the maximum mean discrepancy (MMD)[33]. Low values of MMD imply that all moments of two distributions are matched and thus the entire distribution of perturbed cells is captured in fine detail, beyond the population average (Online Methods provides details). The MMDs between the predicted and observed populations for the

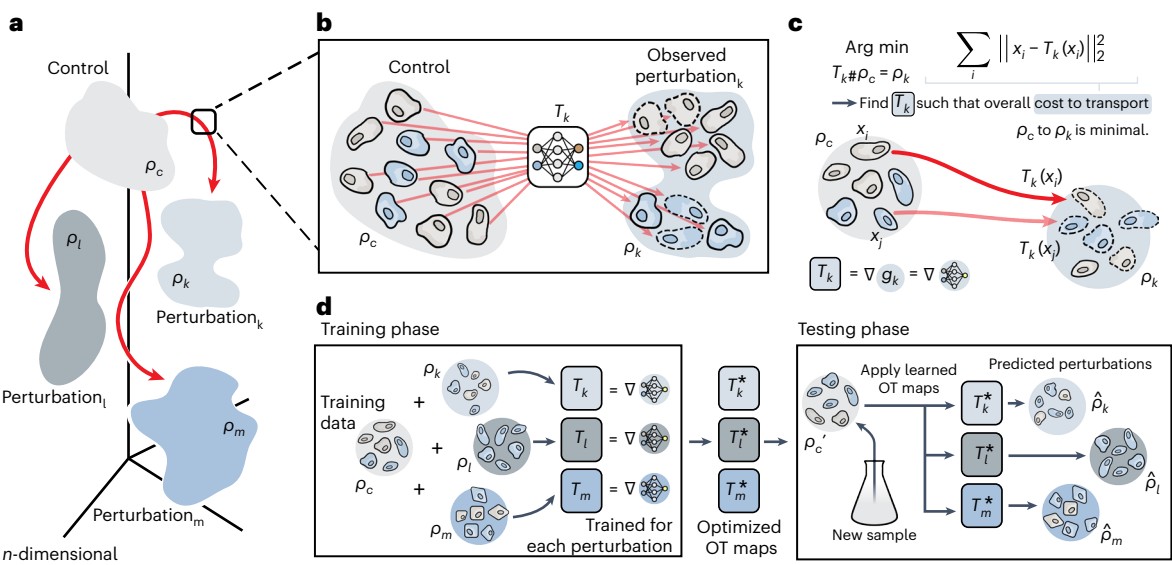

**Fig. 1 | Overview of the CellOT model. a**, Distributions of single cells were measured in either an untreated control state ($\rho_c$) or in one of several perturbed states ($\rho_k$, $\rho_l$, $\rho_m$, …). These distributions lie in a high-dimensional space of profiled features. **b**, For a perturbation $k$, we aim to model it with a function $T_k$ that maps untreated cells in $\rho_c$ to their treated counterparts in $\rho_k$. **c**, Lacking paired measurements, we assume that the perturbation transforms $\rho_c$ into $\rho_k$ under a principle of minimal effort. In particular, we learn $T_k$ using optimal transport theory to directly estimate this distributional mapping as the gradient of the optimal transport dual potential $\nabla g_\theta$. **d**, OT maps are learned for all perturbations independently. Because these maps are fully parameterized, CellOT can be trained, for example, on a set of initially provided samples to then make predictions on untreated cells originating from new, previously unseen samples.

selected perturbations are shown in Fig. 2b,e. For scRNA-seq data, MMD evaluations are computed using the top 50 marker genes. An analysis on the influence of the number of chosen marker genes can be found in Supplementary Fig. 7. In addition to the autoencoder baselines, we include the trivial identity baseline that predicts treatment effects simply by returning the untreated states, as well as a theoretical lower bound, observed, consisting of a different set of observed perturbed cells, thus only varying from the true predictions up to experimental noise. We find that CellOT can approach the lower bound (observed setting), whereas the baseline methods often do not improve much over the identity setting.

Different evaluation metrics across all 35 4i therapies and 6 scRNA-seq therapies are summarized in Supplementary Figs. 5 and 6. Besides MMD, we additionally include the $\ell_2$ mean that measures the distance between the observed and predicted mean drug effect over all features. Lastly, we compare the overall mean correlation coefficient $r^2$ between the predicted and observed data on all features (Online Methods). CellOT outperforms the baselines in both metrics across all treatments, typically by one order of magnitude. We attribute the strong performance of CellOT to its ability to learn a transport function that considers explicitly the data geometries of cell populations through the theory of optimal transport. This hypothesis is supported by the observation that the inter-feature correlation structure remains largely conserved between treated and untreated populations, thus depicting a setting where OT approaches excel. For more information, see Extended Data Fig. 1. Extended Data Fig. 2 visualizes the learned maps, further demonstrating CellOT's ability to model fine-grained responses.

Finally, we computed Uniform Manifold Approximation and Projection (UMAP) projections[34] on a joint set of predicted and observed perturbed cells utilizing the full feature space (Fig. 2c,f). We observe that the perturbed cell states inferred by CellOT are well integrated with the observed perturbed cells. Again, both baselines do not recover the perturbed distribution in its entirety and thus the perturbed state of different subpopulations is not captured consistently.

## CellOT captures cell-to-cell variability in drug responses

Capturing distinct perturbation responses of different cell types within the same sample remains a challenging computational task. To reduce the task's complexity, prediction algorithms can be guided by predefined cell-type labels both in the perturbed and unperturbed states[32] or set to approximate the mean drug response[13]. These simplifications come at a cost: the reliance on a priori knowledge about present and relevant cell types, the assumption that cell types are characterized by the same features before and after a perturbation and that the drug response is uniform within a cell type. In the worst case, these limitations risk masking true and important drug response heterogeneity and thus hamper the discovery of new cell-type- or cell-state-specific perturbation responses (further comparisons are provided in Supplementary Fig. 13). CellOT is free of these limitations and enables scientists to query the predicted single-cell responses at the granularity best suited to answer their biological questions. As a proof of concept, we co-cultured the aforementioned patient-derived melanoma cell lines (Online Methods) at equal ratios and performed a boutique drug screen, during which we exposed cells for 8 h to a panel of 34 drugs and measured the single-cell drug responses with the 4i technology. Using CellOT, we predict the perturbed cell states of a shared set of control (dimethylsulfoxide (DMSO)-treated) cells (Fig. 3a) for each drug. Previous work[7] shows that phosphorylation levels of signaling kinases upon drug treatments are tightly linked to the cellular state. To assess whether this relationship was retained in predicted compared to observed perturbed cells, we analyzed the phosphorylation levels of extracellular signal-regulated kinases (pERK) using the transport maps learned by CellOT on each drug. Using 750 predicted and 750 observed perturbed cells, we computed UMAP projections joint-wise from all features except pERK. Figure 3b shows the predicted and observed population individually annotated with the respective pERK levels of each cell. We found that the spatial organization of the two projections looked almost identical and that pERK levels had a highly comparable distribution across the cells of either class and all drug treatments (further analysis in Extended Data Fig. 3a,b and Online Methods).

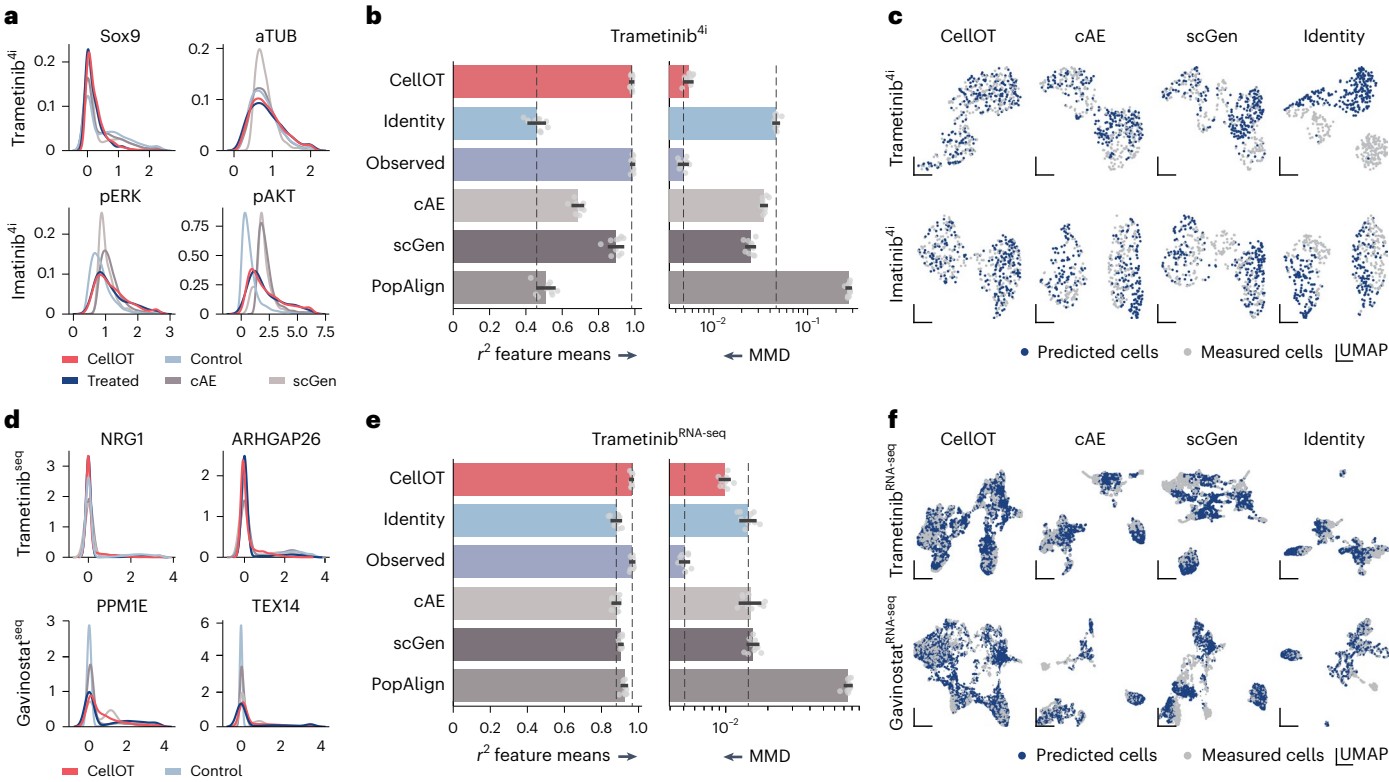

**Fig. 2 | CellOT outperforms current state-of-the-art methods on different data modalities. a–f,** Marginal distribution of marker gene expression (*x* axis) of cells profiled by 4i (**a**) and scRNA (**d**). Observed control and treated states are shown in light and dark blue. CellOT predictions are shown in red and baseline predictions (scGen, cAE and PopAlign) are shown in gray. We compare models based on the distributional distance MMD as well as average correlation coefficient $r^2$ between observed perturbed and predicted perturbed cells, for 4i (**b**) and scRNA (**e**) data. Error bars refer to the standard deviation over ten bootstraps of the test set and the dashed lines correspond to the median of the identity and observed performances. Joint UMAPs of observed treated cells and cells predicted by each model for 4i (**c**) and scRNA (**f**) data. Projections are computed on a joint set of cells, downsampled such that the number of observed perturbed (gray) and predicted perturbed cells (blue) are equal. An identity map compares treated cells to untreated cells. The analysis is conducted for drugs trametinib, imatinib and gavinostat. 4i data were generated using cell lines M130219 and M130429 (Online Methods).

## CellOT disentangles subpopulation-specific drug effects

CellOT allows us to isolate the mode of action of each drug by computing the difference between predicted perturbed cells and untreated control cells. A UMAP embedding of all cells color-coded by the treatment distinctly separates different treatments (Fig. 3c and Extended Data Fig. 3e), all of which CellOT is able to faithfully learn (Supplementary Fig. 5). Such distinct treatment embeddings are not present when accounting only for an average perturbation effect (Extended Data Fig. 3d), indicating the importance of capturing the cellular heterogeneity of drug responses.

Using Leiden clustering on the full feature set, we grouped unperturbed control cells in 12 cellular states (Fig. 3d, Extended Data Fig. 3g and Online Methods). Cellular states 1, 5, 6, 9 and 12 show high levels of MelA and no SOX9 and thus correspond to the melanocytic cell line M130429, whereas the SOX9[+] and MelA[−] states 2, 3, 4, 7, 8, 10 and 11 represent the mesenchymal cell line M130219 (Online Methods). Overall, we find that M130429 cells have higher phosphorylation levels of the measured signaling kinases compared to M130219; a stereotypical spatial organization of cellular states is retained for the majority of the drugs and cell states belonging to the same cell line cluster together (Extended Data Fig. 3f).

Computing the difference between the control and treated state of each drug (the optimal transport cost), allows us to further characterize a drug's severity. Apoptosis inducers (for example, staurosporine), proteasome inhibitors (for example, ixazomig and carfilzomib or the combination treatment carfilzomib + pomalidomide + dexamethasone),

microtubule-stabilizing agents (for example, paclitaxel), c-Met inhibitors (for example, crizotinib) and ATP competitors for multiple tyrosine kinases such as c-KIT and Bcr-Abl (dasatinib) show high transport costs and thus substantial feature changes in all cellular states (Fig. 3e). Other drugs demonstrate less-severe effects in the observed 8-h incubation period. We found that all perturbations increased levels of cleaved caspase 3, an apoptosis marker, in various cellular states and in both cell lines (Extended Data Fig. 3k), with the exception of dasatinib, which specifically induced cell death in cellular states 5, 6, 9 and 19 associated to M130429 (Fig. 3f).

Previous work by Smith et al.[35] reported that M130429 cells reduce metabolic activity upon treatment with inhibitors of MEK (MEKi) and RAF (RAFi), whereas M130219 cells are resistant to these inhibitors. When comparing the responses of the two cell lines to trametinib (MEKi) and MLN2480 (panRAFi) in the MEK and PI3K pathway using pERK and pAKT as the respective readouts, we find that MEKi-sensitive M130429 cells downregulate pAKT and pERK, whereas the MEKi-resistant M130219 cells only downregulate pERK. Consistently, we also found that treatment with MLN2480 results in a similar differential drug response (Extended Data Fig. 3i). This suggests that decoupling of the MEK and PI3K pathways may confer resistance to MEK and Raf inhibitors and constitute an adaptation to the escape of cancer therapy[36]. We found further supporting evidence of pathway crosstalk alteration when we analyzed pAKT and pERK levels upon treatment with a cocktail of trametinib (MEKi) and dabrafenib (BRAFi).

In response to two drugs impinging on the MEK pathway, we observed pERK to be reduced in both cell lines but found increased pAKT levels in the MEKi-resistant cell line M130219 (where resistance was acquired during pre-exposing a patient to MEKi) (Fig. 3f). This finding points toward a compensatory feedback mechanism acquired by M130219 during MEKi treatment by which inhibition of the MEK pathway (quantified as a reduction of pERK) would stimulate signaling through the PI3K pathway, possibly through activation of an upstream receptor kinase[37]. Our results on two co-cultured primary melanoma cell lines treated with various anticancer drugs show that CellOT can accurately capture phenotypic heterogeneity in unperturbed cell populations and predict diverse drug responses by incorporating the underlying cell-to-cell variability without predefined cell line labels.

## CellOT accurately infers cellular responses in unseen patients

The maps between molecular states before and after treatments learned by CellOT contribute to a better understanding of the differences between cells that respond to certain drugs and cells that do not respond. This is crucial for inferring an incoming patient's response to drugs and settings with high cell-to-cell variability. To make predictions on unseen patients, however, we need to demonstrate that the learned maps $T$ model perturbation responses across different patients coherently and robustly, while still predicting personalized treatment outcomes for each patient instead of mere population averages. To test the generalization capacity of CellOT in such an out-of-sample (o.o.s.) scenario, we use a peripheral blood mononuclear cell droplet scRNA-seq dataset. Kang et al.[38] characterize the cell-type specificity and inter-individual variability of the response of eight patients with lupus to interferon (IFN)-β, a potent cytokine that induces genome-scale changes in immune cell transcriptional profiles. In the following, we compare the performance of CellOT and other baselines in an independent-and-identically distributed (i.i.d.) setting, where models see cells from all patients, as well as in the o.o.s. setting, where models do not see cells from a specific holdout patient (Fig. 4a).

As in the previous analysis, we evaluated how accurately CellOT captures the change in the overall expression of different marker genes from control to IFN-β-treated cells and thus how well the predicted gene expression marginals are aligned with the treated population (Fig. 4b). Here, we consider the genes *CXCL11*, *CCL2* and *APOBEC3A*, as they are connected with autoimmune diseases, including systemic lupus erythematosus[39,40] and thus potential therapeutic targets in the management of patients with lupus and, likely, other interferonopathies[39–43]. These selected genes show a large change in expression from the control to the perturbed population, partially exhibiting a bimodal gene expression profile upon perturbation. In contrast to CellOT, the baselines do not accurately predict these large transcriptomic shifts of these genes. An extended analysis of additional genes strongly affected by the IFN-β treatment can be found in Supplementary Figs. 9 and 10.

All models, including CellOT, show little performance drop when modeling the treatment outcome on a new patient using the generalized perturbation model $T_L$ trained on the patient cohort and using the control cells $\rho_{c_z}$ of the unseen patient as input. This becomes evident when comparing the predicted population $\hat{\rho}_{k_z}$ with observations $\rho_{k_z}$ using the MMD metric. Figure 4c displays summary results in which each individual patient was considered for the holdout set. Further evaluation metrics, including the $\ell_2$ feature means, can be found in Supplementary Fig. 8. CellOT outperforms previous baselines both in the i.i.d. and in the o.o.s. setting, while further showing a smaller performance drop when generalizing to the unseen patient. For more results, see Supplementary Fig. 11. These results suggest that the learned optimal transport maps correctly model the shift in the structures of the cellular subpopulation present in all patients, thus robustly performing o.o.s. We repeat the same evaluation for a glioblastoma cohort consisting of seven patients[44]; however, generalization within this setting proved to be difficult for CellOT and all baselines, due to the small size of the cohort and high degree of variance within the responses of each individual. For a complete analysis, see Extended Data Fig. 6.

## CellOT reconstructs innate immune responses across species

The innate immune response is a cell-intrinsic defense program showing high levels of heterogeneity among responding cells, and thus an ideal task for evaluating CellOT's capabilities. Our analysis is based on the dataset collected by Hagai et al.[45] that studies the evolution of innate immunity programs of mononuclear phagocytes within different species, including pigs, rabbits, mice and rats. For this, these primary bone marrow-derived cells are stimulated using LPS. In the following, we test how well CellOT and the baselines reconstruct innate immune responses within species that are not encountered during training. We refer to the generalization task as out-of-distribution (o.o.d.), as unlike the o.o.s. setting, we expect different species to have very distinct responses (Fig. 4d). The holdout set thereby consists of cells derived from either rat or mouse. Extended Data Fig. 4a,b provides an analysis of cross-species similarity and the reasoning behind selecting the holdout set.

Indeed, CellOT accurately reconstructs the innate immune response in both mouse and rat in the i.i.d. and o.o.d. setting. This not only becomes evident through capturing more precisely the mean expression level of marker genes that show high differential expression levels upon addition of LPS, for example, *Nfkb1* (NF-κB), *Oasl1* (Oasl1), *Mmp12* and *Cxcl5* (Fig. 4e and Extended Data Fig. 4c,d), but also through the average correlation coefficient $r^2$ computed between o.o.d. predictions and holdout observations across all genes (Fig. 4f). In particular, CellOT outperforms the baselines when analyzing how well each method captures the heterogeneity of innate immune responses in different species, as demonstrated by low levels of MMD (Fig. 4f). Most notably, our method shows a strong alignment or gene expression marginals of aforementioned marker genes that show complicated bimodal expression profiles upon perturbation (Fig. 4g).

---

**Fig. 3 | CellOT facilitates the multiplexed single-cell characterization of cancer drugs. a**, CellOT training and prediction setup. The 34 CellOT models were trained, one for each drug perturbation. Subsequently, each model was used to predict perturbed cells from a common set of unseen control cells. **b**, UMAP projection constructed with equal numbers of predicted and measured cells from 34 perturbations. Dots correspond to cells, color-coded for measured or predicted pERK intensity. AU, arbitrary unit. **c**, UMAP projection of single-cell perturbation effects using predicted cells. Dots correspond to cells, color-coded for drug treatment (Extended Data Fig. 3 provides the full legend and Online Methods provides the single-cell perturbation effect calculation). **d**, Cell states identified in control cells (Online Methods). Each column represents a cell state. Horizontal axis, cell states sorted based on their association to the cell lines M130219 and M130429. Vertical axis, cellular features (Extended Data Fig. 3 provides the full feature set). The size and hue of the circles are scaled on the feature values. **e**, Clustergram of transport cost (TC) of drug treatments for each cell state (main heat map, blue-yellow color scheme), the sum of TCs (sum) of all states per drug (first column left of the heat map, purple), the coefficient of variation (CV) of TCs per drug (second column left of the heat map, green) and the dendrogram based on the hierarchical clustering the drug's cell state TCs. Cell states are sorted as in **d**. **f**, Cell-state-specific responses to drug treatments. (i) Dasatinib (top). (ii) Trametinib + dabrafenib (bottom). Condition-focused enlargement of UMAP projection from **c** (top left). Same as top left but color-coded for cell-state assignment (top right). Columns represent cell state (cs) and rows show highlighted features (bottom). 'cell-' represents mean cell intensity. Circles are scaled based on drug effect size and the stronger the effect the larger the circles. Negative values are encoded in hues of blue and positive values in red are hues of the respective circles.

**CellOT extends differentiation results to cells of lower potency**

During developmental processes, stem and progenitor cells progress through a hierarchy of fate decisions, marked by a continuous differentiation of cells that refine their identity until reaching a functional end state. By tracking an initial cell population along the differentiation process, CellOT allows us to recover individual molecular cell-fate decisions and developmental trajectories.

Weinreb et al.[46] analyzed the fate potential of hematopoietic stem and progenitor cells, by tracking a broad class of oligopotent and multipotent progenitor cell subpopulations and observing samples on days

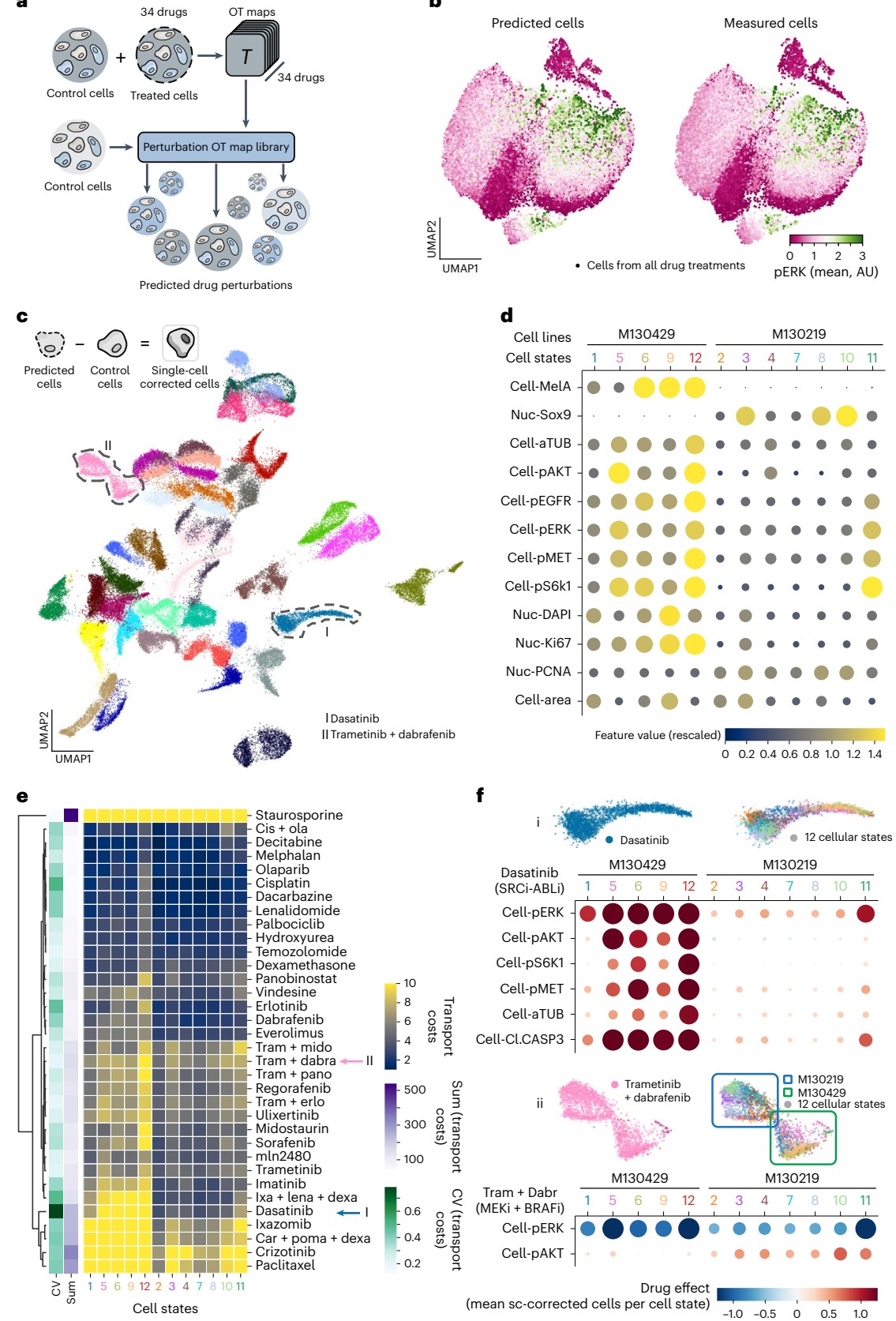

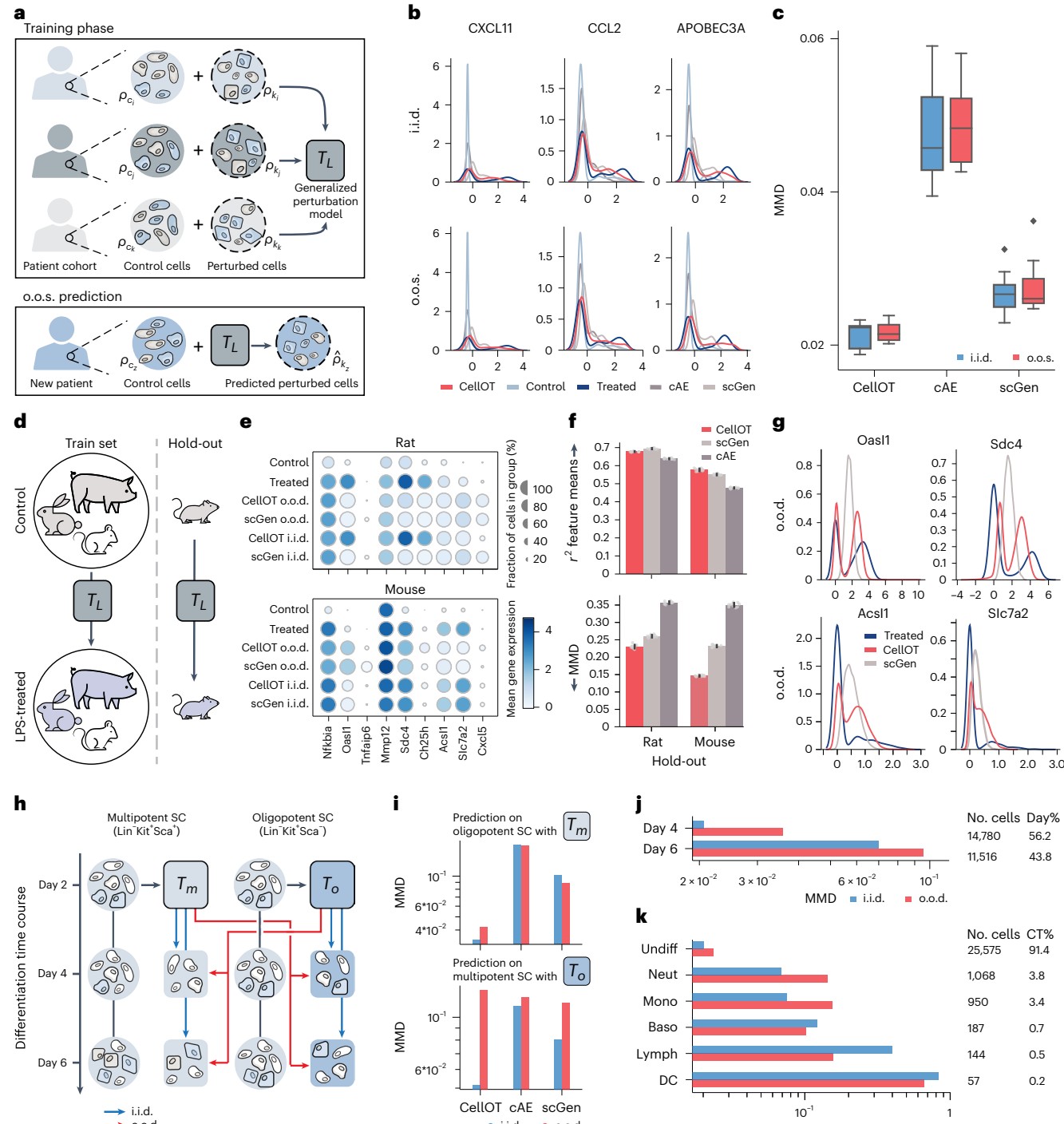

**Fig. 4 | CellOT generalizes to unseen patients and cell subpopulations.**
**a–k**, The o.o.s. (**a–c**) and o.o.d. (**d–k**) setting. **a**, Cells from eight patients with lupus are measured in an untreated and IFN-β-treated state. For each sample, we train two models, an o.o.s. model trained on cells from all other samples and an i.i.d. model trained with additional access to half of the cells in the holdout sample (not shown). **b**, Marginals of predicted cells from the holdout sample in the i.i.d. (top) and o.o.s. (bottom) setting. Predictions for both models are made on the same test set (not used for training the two models). **c**, MMD scores between the predicted distribution and the observed treated distribution across all holdout samples in the i.i.d. and o.o.s. settings. Box plots indicate the median and quartiles. **d**, As an o.o.d. task, we trained CellOT and baselines to predict the response to LPS across different species and test on rat (or mouse) as a holdout species. **e**, Mean gene expression for i.i.d. and o.o.d. predictions for CellOT and scGEN for selected marker genes. **f**, Comparison of o.o.d. performance for $r^2$

correlation feature means and MMD of CellOT and baselines. Data are depicted as the mean ± s.d. across $n = 10$ bootstraps of the test set. **g**, Marginals of the o.o.d. predictions for marker genes showing bimodal expression profiles when using rat as a holdout. **h**, Cells from multipotent and oligopotent subpopulations are measured after 2, 4 and 6 days. We apply CellOT to predict how cells from day 2 develop into the combined set of day 4 and 6, when trained on only multipotent cells ($T_m$) or oligopotent cells ($T_o$). We then apply $T_m$ to predict the o.o.d. oligopotent cells and $T_o$ to predict the o.o.d. multipotent cells. Similar to the o.o.s. setting, i.i.d. models are trained, which includes half of the holdout subpopulation. **i**, MMD scores between the predicted and (observed) developed distributions for all models in both o.o.d. and i.i.d. prediction tasks (jointly for day 4 and 6). Performance of CellOT, when predicting day 4 states (**j**) and day 6 states (**k**) for different cell types in each setting using $T_m$.

2, 4 and 6 (Fig. 4h). Here, we test how well CellOT and other baselines can learn the differentiation process of the cells observed on day 2 to the cells observed on days 4 and 6 (combined) and generalize from one subpopulation to another (o.o.d. setting). We trained two maps, where map $T_o$ was trained exclusively on oligopotent cells and $T_m$ on multipotent cells. The i.i.d. versions of these maps were trained on both oligopotent and multipotent cells, such that each pair of i.i.d. and o.o.d. maps was evaluated on the same test set. Comparing the distributional distance between predicted and observed differentiated cell states using the MMD metric, CellOT outperforms current state-of-the-art methods in this i.i.d. setting for both the oligopotent and the multipotent subsets (Fig. 4i). Furthermore, while baselines struggle to perform in either o.o.d. setting, CellOT is able to generalize its predictions in one direction (from multipotent cells to the oligopotent setting). In contrast to oligopotent cells, multipotent cells have a higher potency and thus can potentially differentiate into more cell types and so we would expect that $T_m$ is more likely to generalize than $T_o$, trained on the less-potent oligopotent cells. When predicting developmental perturbations on multipotent cells using $T_o$, the differentiated cell fates cannot be recovered.

We further compared the performance at different time points and across cell types. Figure 4j shows the accuracy of the modeled development of multipotent cells using map $T_m$ individually for day 4 and day 6 cells, respectively. It is evident that CellOT achieves better results when predicting short-range developmental dynamics instead of states further away in time (further results in Extended Data Fig. 5). This suggests a potential limitation for all of these methods, which might be unable to recover alignments over coarse time resolutions. In addition, while the vast majority of cells on days 4 and 6 were still undifferentiated (undiff), some cells have evolved into neutrophils (neut), monocytes (mono), basophils (baso), lymphoid precursors (lymph) or dendritic cells (DCs). As expected, the performance of CellOT drops in terms of the MMD metric for those cell types that are only sparsely represented in the dataset (Fig. 4k).

## Discussion

In this work we propose CellOT, a framework to model single-cell perturbation responses from unpaired treated and untreated cell states using neural optimal transport. By adequately modeling the nature of the problem through the lens of optimal transport, CellOT determines how perturbations affect cellular properties, reconstructs the most likely trajectory that single cells take upon perturbation and subsequently assists in a better understanding of driving factors of cell-fate decision and cellular evasion mechanisms. CellOT builds on the recent successes of optimal transport applications in single-cell biology[16,17], by introducing a fully parameterized transport map that can be applied to incoming unseen samples. Previous methods[19–21] rely on an unconstrained parameterization of the primal optimal transport map; however, the unconstrained nature of these models makes robust optimization challenging and results in reduced performance[18]. Instead, we learn the transformation of unperturbed to perturbed cell states through the dual optimal transport problem, parameterized via a pair of neural networks constrained to be convex[18]. These constraints are important inductive biases that facilitate learning and result in a reliable and easy-to-train framework, as evidenced by the consistently strong performance of CellOT on several problems without the need for extensive hyperparameter tuning (Online Methods).

CellOT infers the highly complex and nonlinear evolution of cell populations in response to perturbations without making strong simplifying assumptions on the nature of these dynamics. Unlike current approaches comprising autoencoder-based baselines[12–14], CellOT does not necessarily rely on learning meaningful low-dimensional embeddings in which perturbations are modeled as linear shifts. We confirm this advantage through experiments on single-cell responses to different drugs in cancer cell lines obtained with RNA-seq and spatially

resolved 4i measurements, where CellOT consistently outperforms (Fig. 2 and Supplementary Fig. 5). Our evaluations went beyond the often-used average treatment effect and correlation analysis across all cells; we analyzed marginals and computed MMD scores, a strong measure of how well predicted and observed distributions match.

Using CellOT to perform cell-state-aware drug profiling enables us to quantify perturbation effects as a function of the underlying heterogeneity of the studied system, in our case a co-culture of two melanoma cell lines with different sensitivities to drug treatments. In doing so, we sharpen the response profiles of the measured drugs and reveal cell-state-specific responses of multiple signaling pathway in relation to treatment history of the cell line donor. We found that the signaling activity associated with the MEK and PI3K pathways decoupled in cells pre-exposed to MEK inhibitors, a known adaptation mechanism for therapy evasion in melanoma cells[36]. This pathway rewiring is associated with an alteration in the molecular feedback structure of cells from effectors to receptors[36,47]. Thus, combining CellOT with a larger set of combination treatments, multiplexed imaging and cellular systems reflective of disease adaptations may help us to elucidate the molecular mechanisms of signaling pathway evolution in the context of cancer therapy.

We further analyzed how well the learned maps generalize beyond samples used for training (o.o.s. setting) and to different sample compositions (o.o.d. setting). In Fig. 4, we therefore tested CellOT's ability to predict treatment responses in unseen patients with lupus, infer developmental trajectories on stem cells of lower potency and translate innate immune responses across patients. In all cases, CellOT's accuracy and precision were superior to current state-of-the-art methods (Fig. 4). Moreover, the predicted cell states after perturbation are still very close to the actual observed cell states. We consider these results to be particularly promising, as it illustrates that accurate o.o.s. and o.o.d. predictions are indeed possible.

The ability to make o.o.d. predictions, such as on unseen patients, is, however, only feasible if (1) similar samples have been observed in the unperturbed setting and (2) the training set contains cases that are similar not only in their unperturbed state but also their perturbation response. An analysis of patients with glioblastoma treated with panobinostat[44] (Extended Data Fig. 6a–c) confirms this restriction; CellOT and the baselines are able to predict treatment outcomes for those patients that are similar to other patients in both the unperturbed state as well as the perturbation effect (Extended Data Fig. 6f) but fail to capture perturbation effects for patients that exhibit unique responses (Extended Data Fig. 6g). This limitation is important to consider when applying CellOT in o.o.d. settings. To overcome such problems, larger cohorts, additional meta-information and methodological extensions are required. Bunne et al.[48] partially address this issue by deriving a neural optimal transport scheme that can be conditioned on a context, for example, patient metadata, when predicting perturbation responses.

We also observed that the predictive performance for CellOT drops when perturbations are too strong (the cell distributions before and after perturbations are very different) (Fig. 4j); a similar drop was observed for the other methods (Supplementary Fig. 12). The principle underlying the optimal transport theory is ideally suited for acute cellular perturbations during which single cells do not redistribute entirely and randomly in multidimensional measurement space, but typically only in a few dimensions, such that the overall correlation structure is preserved. While this modeling hypothesis is satisfied when perturbation responses are observed via regularly and frequently sampled snapshots, molecular transitions cannot be reconstructed when perturbation responses have progressed too far. For particularly strong or complicated perturbations, cellular multiplex profiles might change too drastically, violating OT assumptions and making it challenging to reconstruct the alignments between unperturbed and perturbed populations based on the minimal effort principle. In such settings, additional information is likely needed, for instance, a model

of the underlying biology or models that integrate observations of multiple smaller time steps.

Despite the stochastic nature of cell-fate decisions and the fact that cellular dynamics are intrinsically noisy[49], CellOT models cell responses as deterministic trajectories. Approaches treating cell-fate decisions as probabilistic events have previously allowed estimation of the full dynamical model to a greater extent than their deterministic counterparts[50]. By connecting OT and stochastic difference equations, recent work[51,52] can build up on CellOT to account for biological heteroscedasticity, at the cost of added model complexity and other simplifying assumptions.

Despite having provided a proof of concept of the capacity of CellOT to model various chemical perturbations for different data modalities through an in-depth analysis of the nature of the learned mapping as well as a demonstration of its versatility in a broad class of applications, CellOT's generalization capacity has been evaluated on relatively small datasets. Crucially, large cohorts consisting of patients with different molecular profiles, such as patients with cancer with various underlying genetics, could result in strongly heterogeneous treatment responses. It is evident that approaches addressing these challenges could readily exploit the upcoming availability of large-scale patient cohort studies. The use of neural optimal transport to learn single-cell drug responses makes for an exciting avenue for future work, including its use to improve our understanding of cell therapies, study drug responses from patient samples and better account for cell-to-cell variability in large-scale drug design efforts.

## Online content

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

## Methods

### Theoretical background

**Optimal transport.** Optimal transport plays dual roles. It induces a mathematically well-characterized distance measure between distributions as well as provides a geometry-based approach to realize couplings between two probability distributions. Let $\mu$ and $\nu$ be two measures in $\mathbb{R}^d$. The optimal transport problem by Monge[53] is defined as

$$\arg \min_{T: T_\sharp \mu = \nu} \mathbb{E}_{X \sim \mu} \parallel X - T(X) \parallel_2^2, \tag{1}$$

where $T$ corresponding to the smallest cost is the optimal transport map. This formulation is non-convex and challenging to solve. Years later Kantorovich[54] provided a relaxation allowing for soft assignments,

$$W(\mu, \nu) = \min_{\gamma \in \Gamma(\mu, \nu)} \mathbb{E}_{(X,Y) \sim \gamma} \parallel X - Y \parallel_2^2, \tag{2}$$

where the polytope $\Gamma(\mu, \nu)$ is $\{\gamma \in \mathbb{R}_+^{n \times m}, \gamma \mathbf{1}_m = \mu, \gamma^\top \mathbf{1}_n = \nu\}$, describes the set of all couplings (or joint distributions) $\gamma$ between $\mu$ and $\nu$. The optimal transport plan $\gamma$ thus corresponds to the coupling between two probability distributions that minimizes the overall transportation cost. Given the OT coupling $\gamma$, the resulting distance $W(\mu, \nu)$ between $\mu$ and $\nu$ is known as the Wasserstein distance. Computing optimal transport distances in (2) involves solving a linear program, and the resulting computational costs are prohibitive for large-scale machine learning problems. Regularizing objective (2) with an entropy term results in significantly more efficient optimization[55] and differentiability w.r.t. its inputs, and thus is commonly used as a loss function in machine learning applications.

Problem (2) denotes the primal formulation of optimal transport. Kantorovich also introduces its corresponding dual[54], which is a constrained concave maximization problem defined as

$$W(\mu, \nu) = \max_{(g,f) \in \Phi_c} \mathbb{E}_\mu[g(x)] + \mathbb{E}_\nu[f(y)], \tag{3}$$

where the set of admissible potentials is $\Phi_c := \left\{ (g,f) \in L^1(\mu) \times L^1(\nu) : g(x) + f(y) \leq \frac{1}{2} \parallel x - y \parallel^2, \forall (x,y) d\mu \otimes d\nu \text{ a.e.} \right\}$[23], Theorem 1.3. Theorem 2.9 by Villani[23] further simplifies the dual problem (3) over the pair of functions $(g,f)$ to

$$W(\mu, \nu) = \underbrace{\frac{1}{2} \mathbb{E} \left[ \parallel x \parallel_2^2 + \parallel y \parallel_2^2 \right]}_{\mathcal{C}_{\mu,\nu}} - \min_{f \in \tilde{\Phi}} \mathbb{E}_\mu[f^*(x)] + \mathbb{E}_\nu[f(y)], \tag{4}$$

where $\tilde{\Phi}$ is the set of all convex functions in $L^1(d\mu) \times L^1(d\nu)$, $L^1(\mu) := \{g$ is measurable & $\int g d\mu < \infty\}$, $f^*(x) = \max_y \langle y, x \rangle - f(y)$ is $f$'s convex conjugate, and the optimal transport map transforming $\mu$ into $\nu$ corresponds to the gradient of $f^*$, i.e., $T = \nabla f^*$. We can recover the optimal transport plan via $\gamma = (\nabla f^* \times \mathrm{Id})_\sharp \mu$. Theorem 2.9 by Villani[23] then proves the existence of an optimal pair $(f, f^*)$ of lower semi-continuous proper conjugate convex functions on $\mathbb{R}^n$ minimizing (3).

**Input-convex neural networks.** Convex spaces such as $\tilde{\Phi}$ in (4), can be parameterized utilizing neural networks which are convex w.r.t. to their inputs. One such parameterization approach comprises so-called input convex neural networks (ICNNs) introduced by Amos et al.[22]. ICNNs are based on fully connected feed-forward networks that ensure convexity by placing constraints on their parameters. An ICNN with parameters $\theta = \{b_i, W_i^z, W_i^x\}$ represents a convex function $f(x; \theta)$ and, for a layer $i = 0 \ldots L - 1$, is defined as

$$h_{i+1} = \sigma_i(W_i^x x + W_i^z h_i + b_i) \text{ and } f(x; \theta) = h_L, \tag{5}$$

where activation functions $\sigma_i$ are convex and non-decreasing, and elements of all $W_i^z$ are constrained to be nonnegative. Despite their

constraints, ICNNs are able to parameterize a rich class of convex functions. In particular, Chen et al.[56] provide a theoretical analysis that any convex function over a convex domain can be approximated in sup norm by an ICNN. Huang et al.[57] further extend ICNNs from fully connected feed-forward neural networks to convolutional neural architectures. Lastly, input convex neural networks have been utilized to parameterize Wasserstein gradient flows[58–60] as well as barycenters[61].

### Model

Recent high-throughput methods provide insight into how cell populations respond to various perturbations on the level of individual cells. Such data, however, is often non-time-resolved and unaligned. Hence, snapshots taken of biological samples before and after perturbations do not provide information on the individual cellular trajectories. Perturbations might include the application of drugs affecting molecular functions in cells, or changes in the cellular environment causing shifts in biological signaling, thus impacting cells and their states in various ways. In the following, we describe our approach, which uncovers single-cell perturbation responses by learning a mapping between control and perturbed cell states. Hereby, let $\mathcal{X}$ denote the biological data space spanned by the measured cell features. We then treat a cell's response to perturbation $k$ as an evolution in a high-dimensional space of cell states $\mathcal{X} = \mathbb{R}^d$.

**Recovering perturbation effects via neural optimal transport.** Given a dataset of $n$ observations $\{x_1^c, \ldots, x_n^c\}, x_i^c \in \mathcal{X}$ drawn from $\rho_c \in \mathcal{P}(\mathcal{X})$, the distribution of cells before applying a perturbation, we aim to predict the distribution of cells $\rho_k \in \mathcal{P}(\mathcal{X})$ upon some perturbation $k$, given a set of separate samples $\{x_1^k, \ldots, x_m^k\}, x_i^k \in \mathcal{X}$.

Perturbation responses of cells are dynamic: After applying perturbation $k$, cell states evolve over time and thus can be modeled as a stochastic process in the cell data space. Despite this time-resolved nature of single-cell responses, we only have access to the distributions of cell states before, $\rho_c$, and after injecting perturbation $k$, $\rho_k$. We thus aim to understand the underlying stochastic process without access to time-resolved perturbation responses by uncovering the map $T$ between $\rho_c$ and $\rho_k$. Given prior biological knowledge, we can assume that perturbations do not drastically or totally alter underlying cellular processes. We thus posit that the evolution of probability distributions of single cells upon perturbation can be modeled via the mathematical theory of optimal transport.

Following Makkuva et al.[18], we thus learn the optimal transport map $T$ (1) between $\rho_c$ and $\rho_k$. Instead of computing a coupling $\gamma$ individually for each pair of cell samples using existing solvers[55] as done by Schiebinger et al.[17], we learn a parameterized optimal transport map using neural networks. The parameterized OT map then serves as a robust predictor for cellular distribution shifts upon perturbations on unseen samples $\{x_i^c\}_{i=1}^{n'} \sim \rho_c$, i.e., such as those of another patient.

**Parametrization of the optimal transport map.** To propose an efficient strategy to learn the optimal transport map, we will build upon celebrated results by Knott[62] and Brenier[63], which relate the optimal solutions for the primal (2) and the dual form (3). As the convex conjugate $f^*$ is very hard to compute, Makkuva et al.[18] propose to approximate $f^*$ in (4) via another convex function $g$, subsequently deriving a max-min formulation over two convex functions, Theorem 3.3 (ref. 18) which reads

$$W(\rho_c, \rho_k) = \max_{\substack{f \in \tilde{\Phi} \\ f^* \in L^1(\rho_k)}} \min_{g \in \tilde{\Phi}} \mathcal{C}_{\rho_c, \rho_k} \underbrace{- \mathbb{E}_{\rho_c}[\langle x, \nabla g(x) \rangle - f(\nabla g(x))] - \mathbb{E}_{\rho_k}[f(y)]}_{\mathcal{V}_{\rho_c, \rho_k}(g,f)}. \tag{6}$$

The intuition behind the approach stems from the fact that

$$\mathbb{E}_{\rho_c}[f^*(x)] = \max_{g \in \tilde{\Phi}} \mathbb{E}_{\rho_c}[\langle x, \nabla g(x) \rangle - f(\nabla g(x))],$$

where we observe that in $\langle x, \nabla g(x) \rangle - f(\nabla g(x)) \leq f^*(x)$ for all functions $g$ the equality is achieved with $g = f^*$[18], Theorem 3.3. In order to learn the resulting optimal transport, i.e., the solution of the minimization problem in (6), Makkuva et al.[18] parameterize both dual variables $g$ and $f$ using input convex neural networks[22], yielding a transport map defined as the gradient of $g$. We then obtain the optimal transport map $T^\star$ via the alternate max-min optimization of

$$(g_\theta^\star, f_\phi^\star) \leftarrow \arg\max_\phi \min_\theta \mathcal{C}_{\rho_c, \rho_k} - \mathcal{V}_{\rho_c, \rho_k}(g_\theta, f_\phi), \tag{7}$$

where $T^\star = \nabla g_\theta^\star$ and $\theta$ and $\phi$ are the parameters of each ICNN.

**Predicting perturbation effects via CellOT.** The framework described above allows us to recover maps between control $\{x_1^c, \ldots, x_n^c\}$ and perturbed cells $\{x_1^k, \ldots, x_m^k\}$, giving insights into cellular response trajectories upon application of a perturbation $k$. Given a set of perturbations $K$, and sample access to the control distribution $\rho_c$ as well as distributions $\rho_k$ for each perturbation $k \in K$, CellOT learns the optimal pair of dual potentials $(g_{\theta_k^\star}, f_{\phi_k^\star})$ by solving (7). Given parametrizations of the convex potentials for each $k$, CellOT then predicts the transformation of a control cell $x_i^c$ upon perturbation $k$ via $\hat{x}_i^k = \nabla g_{\theta_k^\star}(x_i^c)$, i.e., samples following the predicted perturbed distribution $\hat{\rho}_k = (\nabla g_{\theta_k^\star})_\# \rho_c$. CellOT thus provides a general approach to predict state trajectories on a single-cell level, as well as understand how heterogeneous subpopulation structures evolve under the impact of external factors.

**Neural optimal transport.** Beyond the chosen approach, other efforts have investigated ICNN-based approaches as fast and scalable approximations to (1). Taghvaei et al.[64] consider solving (4) by parameterizing $f$ with an ICNN and solving for $f^*$ at each step, which, however, is computationally very expensive. Makkuva et al.[18], and as such the approach considered in this work, extend this work by approximating $f^*$ with another ICNN $g$ transforming the problem into a max-min optimization of two input convex neural networks (see (7)). Lastly, Huang et al.[57] introduce a novel, OT-inspired parameterization of normalizing flows utilizing ICNNs. See Korotin et al.[65] for a detailed comparison of the current state of neural optimal transport solvers.

**Limitations.** While single-cell expression profiling provides a detailed look into the molecular states of individual cells, these observations are often destructive and thus does not allow for continuous measurements of molecular properties over time. There have been numerous proposals for methods to uncover the dynamics of individual cells from population data, but all of them face the same challenge: sequentially observed distribution of cell states can be produced by multiple dynamics and mechanisms of gene regulation. The ill-defined nature of the problem makes it necessary to pose certain assumptions on the underlying cellular dynamics.

The mathematical foundation of this work builds on the biological intuition that perturbations incrementally alter the molecular profiles of cells. This principle aligns with the theory of optimal transport and, following previous work[17], serves naturally as the model foundation of CellOT. If this principle is violated, however, and perturbations strongly disrupt the population to an unidentifiable level, the performance of CellOT as well as other methods drops (see Discussion). In these instances, a more complicated mathematical machinery would be needed. Such tools, however, are currently unable to scale to settings with more than a few genes[66]. Thus, we rely on a fine granularity of the time course to recover large cell state changes between consecutive time points[67].

Furthermore, if a system exhibits rotations and oscillations within two consecutive snapshots not captured by measurements, models based on optimal transport as well as previous tools[68] will not be able to recover such complex dynamics. This is in part also due to the current choice of the cost function, which, due to theoretical constraints and practical performance, is set to the Euclidean distance (2). We leave it to future work, to investigate choices of alternative cost functions.

Beyond, the current system is not able to recover effects (other than cell flux) that change the distribution of cells between time points, for example, proliferation and death[67]. Recent works, however, propose extensions to the classical neural optimal transport scheme that account for cell death and birth[69].

Lastly, current developments in bioengineering aim at overcoming the technological limitation of destructive cell assays. Chen et al.[70] propose a transcriptome profiling approach that preserves cell viability. Weinreb et al.[46] capture cell differentiation processes while clonally connecting cells and their progenitors through barcodes. These methods thus offer (lower-throughput) insights that provide individual trajectories of cells over time, i.e., an alignment between distinct measurement snapshots. Somnath et al.[52] propose a novel algorithmic framework connected to optimal transport that is able to make use of such (partially) aligned datasets[71,72].

## Datasets and preprocessing

**Single-cell multiplex data.** Biologists have various powerful technologies at their disposal, capable of capturing multivariate single-cell measurements. High-content imaging, particularly when augmented by multiplexing abilities such as by Iterative Indirect Immunofluorescence Imaging (4i)[5], is ideally suited to study heterogeneous cell responses. With 4i, fluorescently labeled antibodies are iteratively hybridized, imaged, and removed from a sample to measure the abundance and localization of proteins and their modifications. Thus, 4i quickly generates large, spatially resolved phenotypic datasets rich in molecular information from thousands of treated and untreated (control) cells. Additionally to the multiplexed information generated by 4i, cellular and nuclear morphology are routinely extracted from microscopy images (without the need for 4i) by image analysis algorithms[73].

The cells were seeded in a 384-well plate, and allowed to settle and adhere overnight. Drugs and dimethyl sulfoxide as the vehicle control was added to the cells the next morning and incubated for 8 hours, after which the cells were fixed with paraformaldehyde. Subsequently, 6 cycles of 4i were performed, for which the images were acquired with an automated high-content microscope. We utilized a mixture of two melanoma tumor cell lines (ratio 1:1) in order to image a total of 97,748. For this, we consider two co-cultured primary melanoma cell lines (M130219 and M130429), which were derived from the same melanoma patient from different body sites. M130219 originates from a subcutaneous biopsy taken during treatment with Bimetinib (MEKi), whereas M130429 was derived from a bone autopsy one month after stopping said targeted therapy[30]. Both cell lines share the same driver mutation (NRAS Q61R) but are phenotypically diverse. Consequently, the cell lines are also classed as two different melanoma subtypes due to, amongst others, differences in marker expression[30]: the former a mesenchymal subtype (SOX9+, MelA-), the latter a melanocytic subtype (Sox9-, MelA+). 10,995 cells are imaged in the DMSO-treated control state and the rest are treated with one of 34 cancer therapies. Between 2,000 and 3,000 cells are profiled per treatment.

All image analysis steps were performed by our in-house platform called TissueMAPS (https://github.com/TissueMAPS). The steps included illumination correction[74], alignment of images from different acquisition cycles using Fast Fourier Transform[75], segmentation of nuclei and cell outlines[76], as well cellular and nuclear measurements of intensity and morphology features using the `scikit-image` library[77].

The extracted marker intensities and morphological features are then re-normalized to the same numerical scale by dividing each feature with its 75th percentile computed on control cells. Values are then transformed with a log1p ($x \leftarrow \log(x + 1)$) function. A total of 47 features are reported, 21 morphological features and 26 protein intensities.

**Single-cell RNA sequencing data.** For the statefate[46] and SciPlex 3[31] datasets, raw counts were obtained from their GSA uploads. For each, the `scanpy` toolbox[78] was used to perform library size normalization, cell and gene filtering, and a log1p transformation. For all datasets, we consider the 1,000 highly-variable genes, which were computed based on the training set only. Highly variable genes are thereby computed using the `scanpy`'s `highly_variable_genes` function. Preprocessing for the lupus patients[38] and cross-species dataset[45] were inherited from Rybakov et al.[79] and Lotfollahi et al.[13], and we would like to thank the author for hosting this dataset. Lastly, the preprocessing of the glioblastoma patient dataset[44] was adapted from Peidli et al.[80]. See the Data Availability section for further details.

### Training and technical details
**Setup.** In the i.i.d. setting, we split all cell datasets into train, test, and evaluation set, where the test and evaluation set consists of 500 to 1,000 cells, dependent on the size of the original dataset. The split is performed on each drug and control condition independently. In the o.o.d. setting, the model does not have access to cells of the holdout condition. To provide a fair comparison at the point of model deployment, we evaluate both i.i.d. and o.o.d. models on the same subset of holdout cells. Different from models trained o.o.d., i.i.d. models have seen the remaining holdout cells during training. At evaluation time, we use the same set of cells to ensure that results are comparable.

For scRNAseq datasets, we select hyperparameters for autoencoder models by doing a grid search over parameters summarized in Supplementary Table 1 and selecting the configuration that has the smallest reconstruction error over non-zero features. scRNA-seq datasets comprise more than 1,000 differentially expressed genes, typically assumed to lie in a low-dimensional manifold arising from the constraints of the underlying gene regulatory networks[12,13]. When applying CellOT to scRNA data, we use the same encoder that is used for SCGEN and embed gene expression data into a 50-dimensional latent space before applying CellOT. All models are trained for 250k iterations.

**CellOT network architecture.** As suggested by[18], we relax the convexity constraint on $g_\theta$ and instead penalize its negative weights $W_l^z \in \theta$

$$R(\theta) = \sum_{W_l^z \in \theta} \left\| \max\left(-\overset{z}{W}_l, 0\right) \right\|_F^2. \tag{8}$$

The convexity constraint on $f_\phi$ is enforced after each update by setting the negative weights of all $W_l^z \in \phi$ to zero. Thus the full training objective is

$$\max_{\phi:W_l^z \geq 0, \forall l} \min_\theta f_\phi(\nabla g_\theta(x)) - \langle x, \nabla g_\theta(x)\rangle - f_\phi(y) + \lambda R(\theta). \tag{9}$$

**Hyperparameters.** To learn the optimal transport maps, we use a batch size of 256, an ICNN architecture of 4 hidden layers of width 64, a learning rate of 0.0001 ($\beta_1 = 0.5, \beta_2 = 0.9$), and $\lambda = 1$. $f$ and $g$ are learned in an iterative fashion. $f$ is updated by fixing $g$ and maximizing (9) with a single iteration. For each iteration, $f$ is then fixed, and an inner loop of 10 updates is run to minimize $g$. To train all networks, we use the Adam optimizer[81]. For all data modalities, i.e., different tasks involving 4i or scRNA-seq data, the selection of hyperparameters remains the same. Hyperparameters for the autoencoder-based baselines, however, were selected based on a grid search over parameters listed in Supplementary Table 1, which also contains the final choice. When comparing cAE to its variational counterpart (cVAE), we found no meaningful differences between the representations learned by either model. In practice, the weight of the KL term in the VAE is chosen such that the likelihood component of the loss is orders of magnitude larger. Furthermore, one of the main features of the variational counterpart, i.e., the ability to generate new cell states by sampling from the prior

distribution, is usually not utilized. An extended discussion on the baselines and related work can be found in Supplementary Section A.

### Evaluation
**Metrics.** Since we lack access to the ground truth set of control and treatment observations on the single-cell level, we analyze the effectiveness of CellOT using evaluations that operate on the level of the distribution of real and predicted perturbation states. Three metrics are considered, i.e., MMD, $\ell_2$ feature means, and the average correlation coefficient $r^2$ of the feature means.

$\ell_2$ feature means refers to the $\ell_2$-distance between means of the observed and predicted distributions. Similarly, $r_2$ feature means refers to the correlation of the means of the observed and predicted distributions. However, metrics based only on feature means can be insensitive in settings where crucial heterogeneity is not captured. Consider, for example, a target distribution with multiple modes. These metrics will favorably evaluate a uni-modal predicted distribution that simply models the mean of this multi-modal distribution. To this end, we include a distributional distance sensitive to this type of behavior by measuring differences in the properties of higher moments, i.e., the maximum mean discrepancy.

MMD refers to the kernel maximum mean discrepancy[33], a metric to measure distances between distributions. Given two random variables x and y with distributions p and q, and a kernel function $\phi$, Gretton et al.[33] define the squared MMD as

$$\text{MMD}(p, q; \phi) = \mathbb{E}_{x,x'}[\phi(x, x')] + \mathbb{E}_{y,y'}[\phi(y, y')] - 2\mathbb{E}_{x,y}[\phi(x, y)].$$

We report an unbiased estimate of $\text{MMD}(r_k, \hat{r}_k)$ where the expectations are evaluated by averages over the cells in each set. The RBF kernel is employed, and as is usually done, reports the MMD as an average over several length scales, i.e., `np.logspace(1, -3)`.

Lastly and aligned with previous works, we report the overall average correlation coefficient $r^2$ between predictions and observations.

**Feature selection.** For 4i data[5], the above metrics are computed using all 47 features. The $\ell_2$ feature means and the average correlation coefficient $r^2$ are also computed on the entire gene set, i.e., ~1,000 genes for scRNA-seq or ~50 features for 4i data. Due to the high dimensionality of scRNA data, we report the MMD using the top 50 marker genes. Marker genes are computed for each perturbation with the `scanpy`[78] function `rank_genes_groups`, using the untreated control cells as reference. The influence on the number of selected marker genes is further analyzed in Supplementary Fig. 7, exemplary on Trametinib for the SciPlex 3 dataset[31]. The analysis thereby demonstrated that the MMD computation is biased with increasing dimensionality.

### Reporting summary
Further information on research design is available in the Nature Portfolio Reporting Summary linked to this article.

## Data availability
Raw published data for the SciPlex 3 (ref. 31), patients with lupus[38], patients with glioblastoma[44] and statefate dataset[46] are available from the Gene Expression Omnibus under accession codes GSM4150378, GSE96583, GSE148842 and GSE140802, respectively. Data from the cross-species dataset[45] are hosted on the BioStudies database of EMBL-EBI under code E-MTAB-6754. A full set of links can be found in that publication. The processed datasets of all tasks can be accessed at https://doi.org/10.3929/ethz-b-000609681. Source data are provided with this paper.

## Code availability
CellOT is written Python and uses standard Python libraries. The CellOT library is available at https://doi.org/10.3929/ethz-b-000612005.

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

## Acknowledgements

We are grateful to H.Y. Che and X. Bonilla for their helpful comments, corrections and discussions. C.B. and A.K. received funding from the Swiss National Science Foundation under the National Center of Competence in Research Catalysis under grant agreement 51NF40 180544. L.P. is supported by the European Research Council (ERC-2019-AdG-885579), the Swiss National Science Foundation (grant 310030_192622), the Chan Zuckerberg Initiative and the University of Zurich. G.G. received funding from the Swiss National Science Foundation and InnoSuisse as part of the BRIDGE program, as well as from the University of Zurich through the BioEntrepreneur Fellowship. K.L. and S.G.S. were partially funded by ETH Zürich core funding (to G.R.) and from the Tumor Profiler Initiative (to G.R.). K.L. was also partially funded by the funding programme Cancer Center Cologne Essen of the Ministry of Culture and Science of the State of North Rhine-Westphalia.

## Author contributions

G.G., K.L., L.P., A.K. and G.R. conceptualized the study. C.B. developed CellOT, C.B. and S.G.S. implemented CellOT and S.G.S. and C.B. performed CellOT experiments. G.G. and J.S.C. planned and performed the 4i experiments. S.G.S., C.B., G.G., K.L., L.P., A.K. and G.R. analyzed results and generated figures. C.B., S.G.S., G.G., K.L., L.P., A.K. and G.R. wrote the paper. K.L., L.P., A.K. and G.R. supervised the study. M.L. provided reagents and cell lines and gave feedback on the paper and expert advice on the cell lines.

## Funding

## Competing interests

G.G. and L.P. have filed a patent on the 4i technology (patent WO2019 207004A1). The other authors declare no competing interests.

## Additional information

**Extended data** is available for this paper at https://doi.org/10.1038/s41592-023-01969-x.

**Correspondence and requests for materials** should be addressed to Kjong-Van Lehmann, Lucas Pelkmans, Andreas Krause or Gunnar Rätsch.

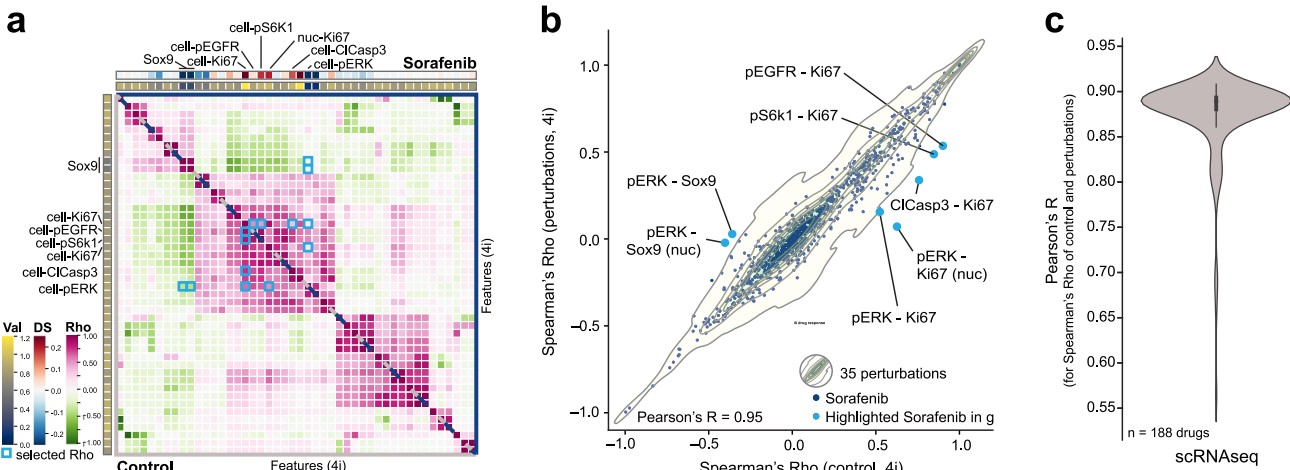

**Extended Data Fig. 1 | Analysis of dataset structures for the 4i and SciPlex 3 dataset.** Analysis of dataset structures for the 4i dataset (**a**, **b**) and the SciPlex 3 dataset (**c**). **a**, Spearman correlations between all feature pairs computed in 4i control cells (bottom) and Sorafenib-treated cells (top). Row colors show the mean value of each feature in control cells and column colors show the effect of the drug on each feature as computed by the difference in means between control and treatment. Correlations that changed the most under the perturbation are highlighted in blue. **b**, Density plot of feature correlations in the control setting vs. treated setting for all 35 4i treatments. Sorafenib values (corresponding to elements in **b** are scattered above and light blue points correspond to blue boxes in **b**. **c**, Feature correlation between in the control setting vs. treated setting for all cancer drugs of the SciPlex 3 dataset.

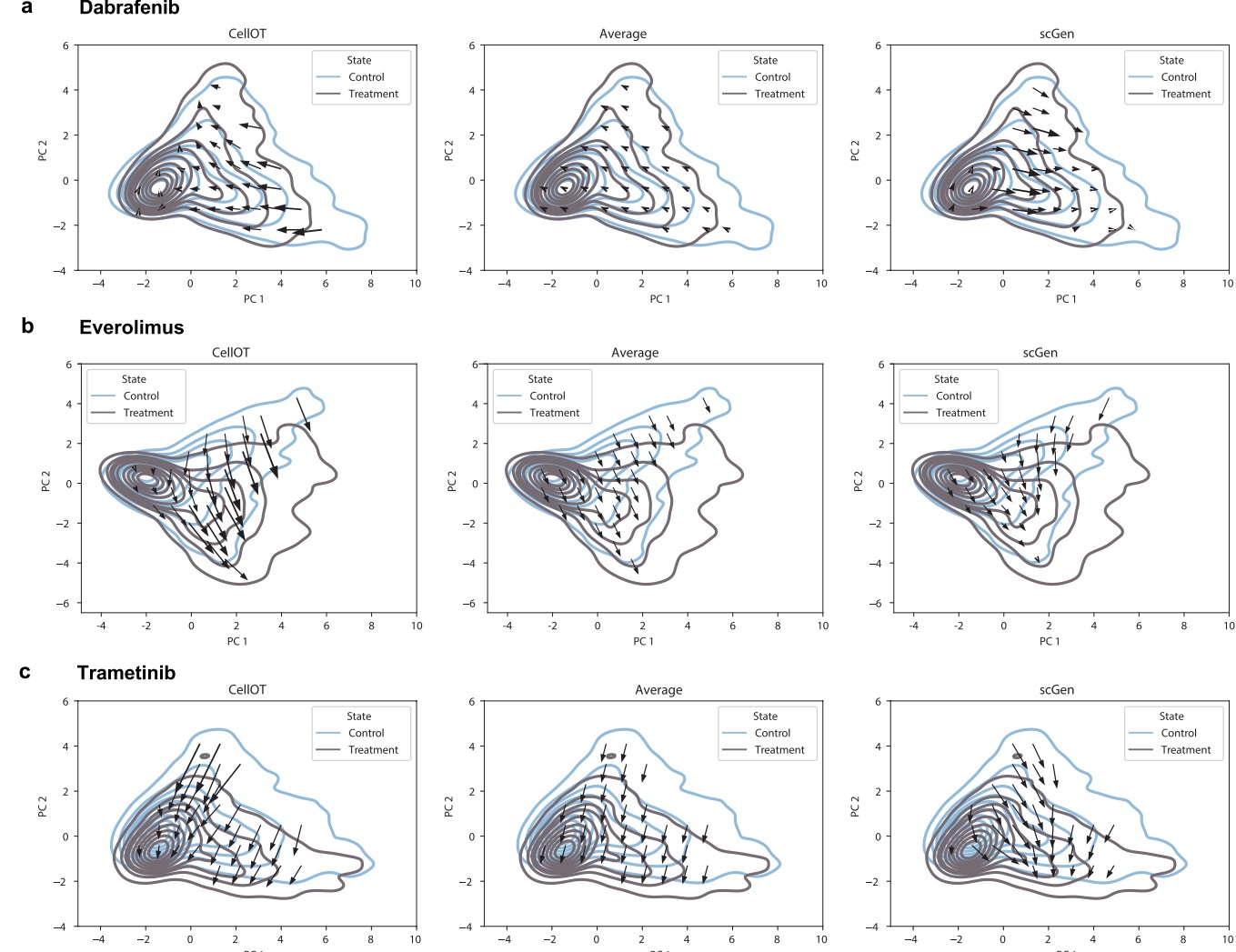

**Extended Data Fig. 2 | Visualization of the learned vector field.**
The perturbation response on the single-cell level is described for **a**, Dabrafenib, **b**, Everolimus and **c**, Trametinib of the 4i dataset for CellOT, the average effect and scGen on the first two principal components. Cellular responses are computed as the predicted treated state minus the observed control state for each individual cell. Arrow tails are placed in a grid within PC space and arrowheads correspond to the average response of cells within each neighborhood, projected into PC space.

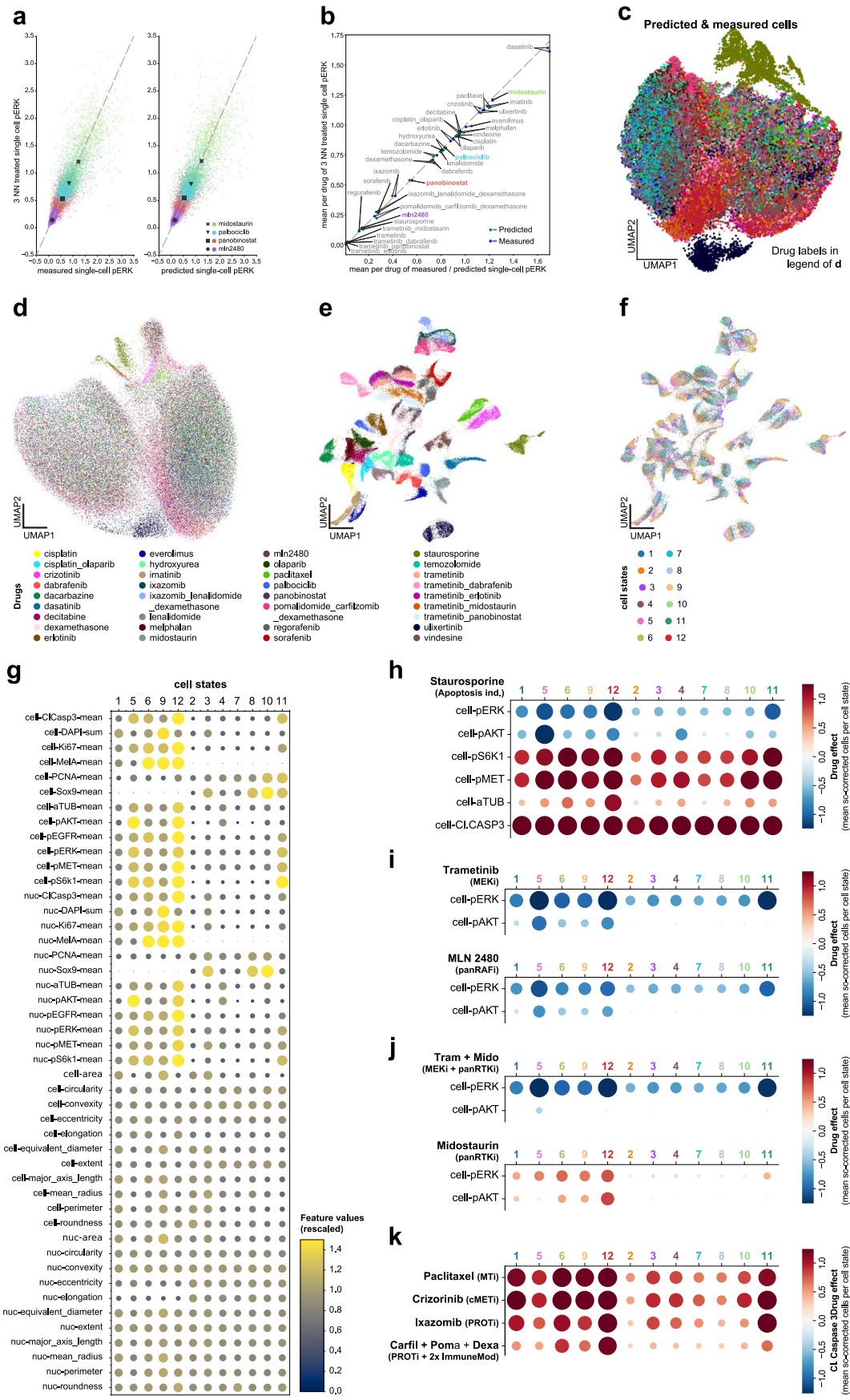

**Extended Data Fig. 3 | See next page for caption.**

**Extended Data Fig. 3 | Extended analysis of CellOT-predicted responses to 34 cancer treatments. a**, High similarity of measured and CellOT-predicted single-cell pERK (phosphor ERK1/2) values at the single-cell level. Scatter plots compare the relationship between measured pERK values of cells (left) treated with Midostaurin (green dots), Palbociclib (blue dots), Panobinostat (red dots) and MLN2480 (purple dots) or (right) predicted for those drugs along the horizontal axis to their corresponding 3NN cells on the vertical axis. For details on the generation of 3NN data, see Online Methods. X mark, square, inverted triangle and circle represent the mean of the respective measurements per drug. The dashed gray line indicates the diagonal along which the measurements would correlate perfectly. **b**, The high similarity of measured and CellOT-predicted single-cell pERK (phosphor ERK1/2) values at the population level across all drug perturbations. Drug average of measured (blue dots) and predicted (green dots) pERK values compared to their respective 3NN measurement (see Online Methods). Drug treatments highlighted in color correspond the those presented in panel **a**. The dashed gray line indicates the diagonal along which the measurements would correlate perfectly. **c**, Projection of measured perturbed and predicted perturbed cells in a shared UMAP space. Each cell is color-coded according to the perturbation from which it originates. **d**, Projection of mean-corrected measured perturbed cells in a UMAP space. Each cell is color-coded according to the perturbation from which it originates. Mean correction was achieved by subtracting calculating the mean of every feature for all cells in the control condition and subtracting the calculated feature means

from the feature values of individual cells. **e**, Projection of single-cell corrected, predicted perturbed cells in a UMAP space. Each cell is color-coded according to the perturbation model with which it was predicted. See Online Methods for details on the single-cell correction. **f**, Projection of single-cell corrected, predicted perturbed cells in a UMAP space. Each cell is color-coded according to its assignment to one of the 12 cell states. See Online Methods for details on cell state assignment. **g**, Feature value overview of the 12 identified cell states in DMSO-treated (Control) cells. Each column represents a cell state and each row a feature. Circles are colored and scaled based on feature value, from small size in blue for low feature values, to large circles in yellow for high feature values. **h-j**, Drug effect overview of the 12 identified cell states in **h**, Staurosporine (apoptosis ind.m apoptosis inducer, **i**, Trametinib (MEKi, MEK inhibitor), MLN2480 (panRAFi, panRAF inhibitor), **j**, Trametinib + Midostaurin (Tram + Mido, MEK inhibitor + pan Receptor Tyrosine Kinase inhibitor (panRTK)), Midostaurin (panRTK). Each column represents a cell state and rows represent features. 'cell-' stands for mean cell intensity. Circles are scaled based on drug effect, the larger the ± effect the larger the circles. Negative values are encoded in hues of blue and positive values in red hues of the respective circles. **k**, Effect of drug treatments on levels of cleaved Caspase 3 (cleaved Caspase 3) in the 12 identified cell states. Each column represents a cell state, each row a drug treatment. Circles are scaled based on drug effect, the larger the ± effect the larger the circles. Negative values are encoded in hues of blue, positive values in red hues of the respective circles.

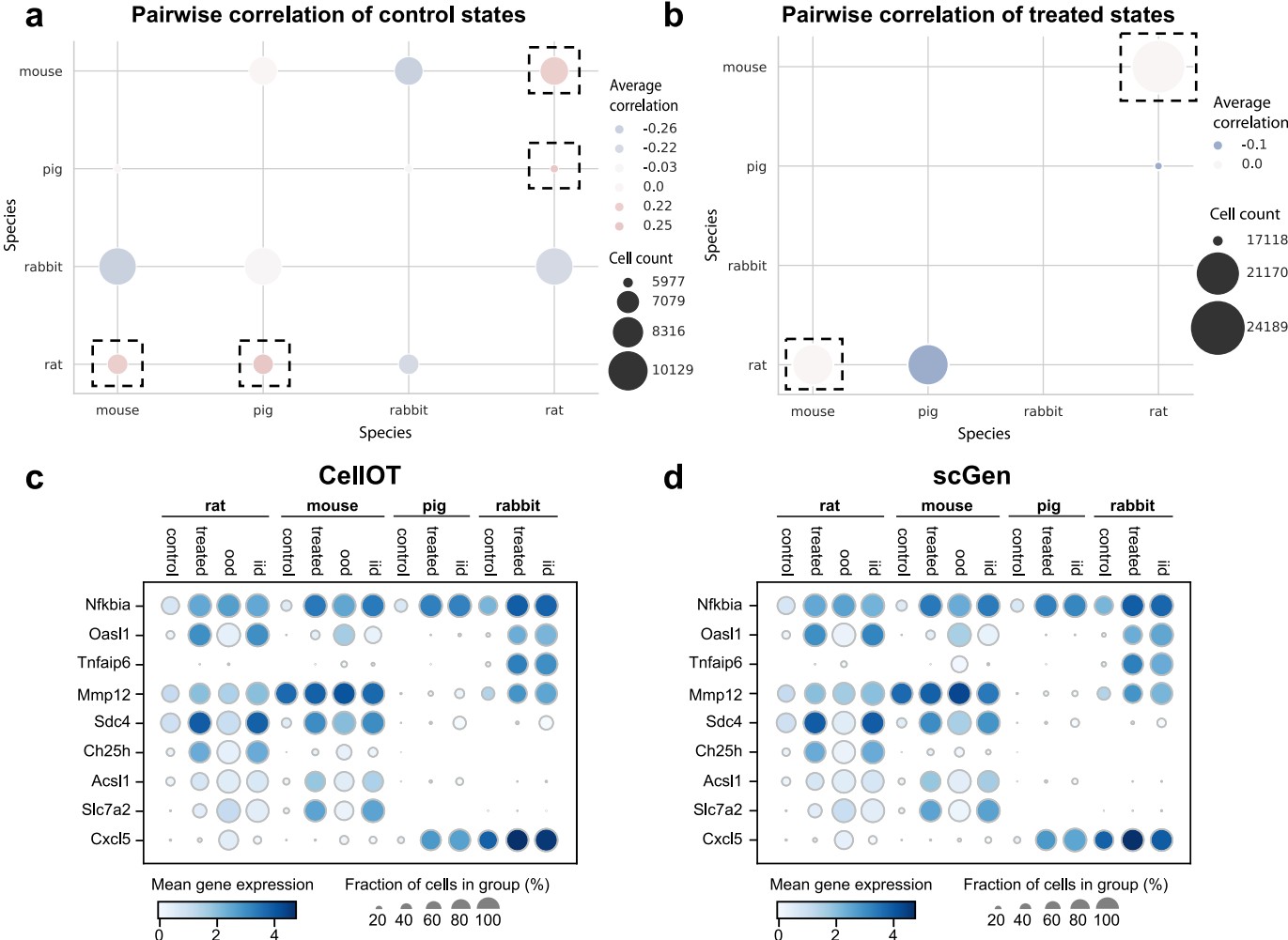

**Extended Data Fig. 4 | Extended analysis of the cross species dataset.**
**a**, Pairwise average correlation of the PCA embeddings of the control states
between species. **b**, Pairwise average correlation of the PCA embeddings of the
treated states between patients, masked to only those patient pairs that showed
a positive correlation in the control states. Only rat and mouse show consistent

responses, that is, a positive correlation of the control states and non-negative
correlation of the respective target cells and are thus chosen for the o.o.d.
analysis. I.i.d. and o.o.d. results measured in the average gene expression for both
**c**, CellOT and **d**, scGen.

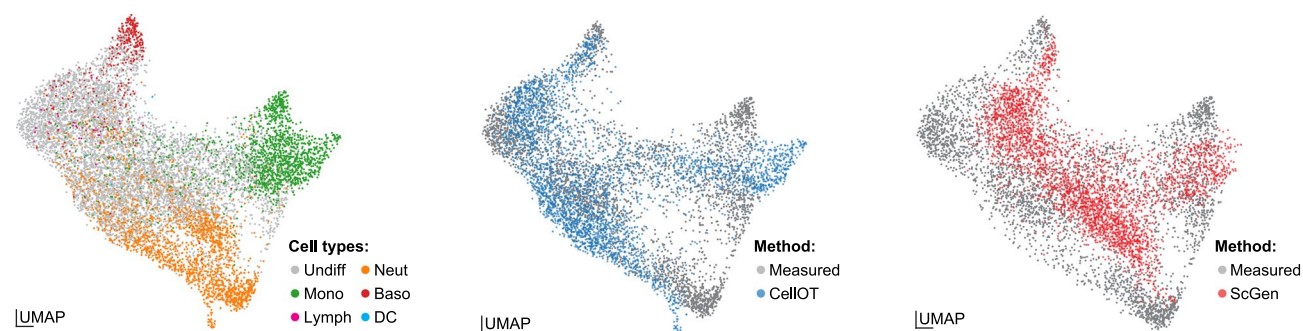

**Extended Data Fig. 5 | Integration of predicted and observed perturbed state in the statefate dataset.** Joint UMAP projections are computed for observed, CellOT and scGen predictions. In each axis, projections are colored by cell type (left), CellOT predictions (middle) and scGen predictions (right). For both our method, CellOT and the baseline scGen, the UMAP highlights the observed and predicted perturbed cell states.

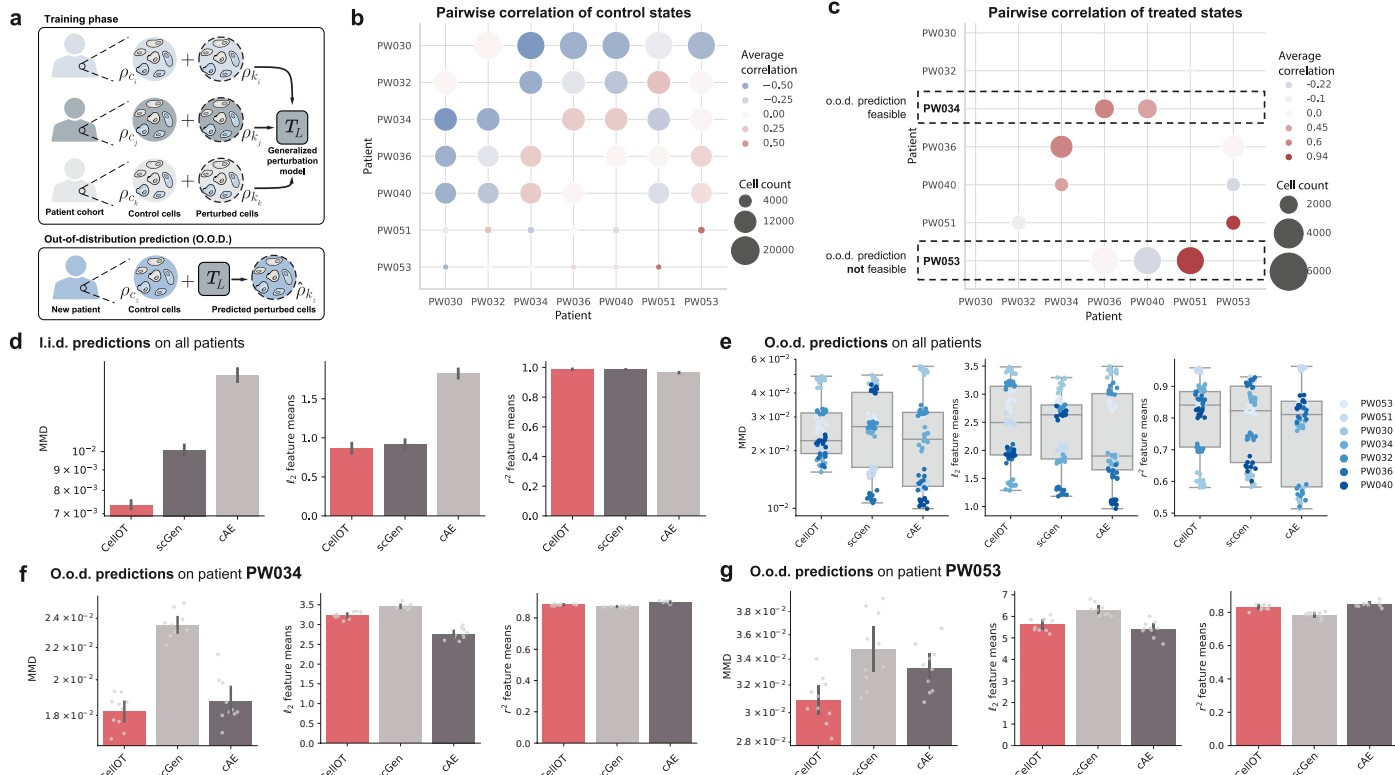

**Extended Data Fig. 6 | Analysis and results of the glioblastoma dataset consisting of seven patients. a**, Cells from seven glioblastoma patients are measured in an untreated and Panobinostat-treated state. For each sample, we train two models, an o.o.d. model trained on cells from all other samples but the holdout patient we test on and an i.i.d. model trained with additional access to half of the cells in the holdout sample. **b**, Pairwise average correlation of the PCA embeddings of the control states between patients. **c**, Pairwise average correlation of the PCA embeddings of the treated states between patients, masked to only those patient pairs that showed a positive correlation in the control states. Only patient PW034 positively correlates with all other patients.

Other patients, such as PW053, correlate and anti-correlate with other patients in the treated state. Performance comparison between CellOT and baselines for different metrics in the **d**, i.i.d. setting (mean standard deviation across 7 samples, 10 bootstraps of the test set per sample), **e**, o.o.d. setting for all patients (box plots show median, minima and maxima), **f**, o.o.d. setting for a patient positively correlating with all patients that are also similar in the control state, **g**, o.o.d. setting for a patient where similar patients in the control state show different responses (correlation and anti-correlation) in the treated states. Data in f and g are presented as the mean +/- standard deviation across n=10 bootstraps of the test set.

# nature research

# Reporting Summary

Nature Research wishes to improve the reproducibility of the work that we publish. This form provides structure for consistency and transparency in reporting. For further information on Nature Research policies, see our Editorial Policies and the Editorial Policy Checklist.

## Statistics

For all statistical analyses, confirm that the following items are present in the figure legend, table legend, main text, or Methods section.

| n/a | Confirmed |
|-----|-----------|
| ☒ | ☐ The exact sample size (*n*) for each experimental group/condition, given as a discrete number and unit of measurement |
| ☒ | ☐ A statement on whether measurements were taken from distinct samples or whether the same sample was measured repeatedly |
| ☒ | ☐ The statistical test(s) used AND whether they are one- or two-sided *Only common tests should be described solely by name; describe more complex techniques in the Methods section.* |
| ☒ | ☐ A description of all covariates tested |
| ☒ | ☐ A description of any assumptions or corrections, such as tests of normality and adjustment for multiple comparisons |
| ☒ | ☐ A full description of the statistical parameters including central tendency (e.g. means) or other basic estimates (e.g. regression coefficient) AND variation (e.g. standard deviation) or associated estimates of uncertainty (e.g. confidence intervals) |
| ☒ | ☐ For null hypothesis testing, the test statistic (e.g. *F*, *t*, *r*) with confidence intervals, effect sizes, degrees of freedom and *P* value noted *Give P values as exact values whenever suitable.* |
| ☒ | ☐ For Bayesian analysis, information on the choice of priors and Markov chain Monte Carlo settings |
| ☒ | ☐ For hierarchical and complex designs, identification of the appropriate level for tests and full reporting of outcomes |
| ☒ | ☐ Estimates of effect sizes (e.g. Cohen's *d*, Pearson's *r*), indicating how they were calculated |

*Our web collection on statistics for biologists contains articles on many of the points above.*

## Software and code

Policy information about availability of computer code

| | |
|-----|-----------|
| Data collection | A detailed description of the 4i related software that was used to collect marker and other cell associated information is described in the Online Methods section. Moreover, the preprocessing of the scRNA data is also described in the Online Methods section. |
| Data analysis | This work presents a new analysis method, described in detail in the Online Methods section. The code is available online (see Code Availability statement). Additional details on in-vitro and in-silico experiments can be found in Section S4 of the Supplementary Information. |

For manuscripts utilizing custom algorithms or software that are central to the research but not yet described in published literature, software must be made available to editors and reviewers. We strongly encourage code deposition in a community repository (e.g. GitHub). See the Nature Research guidelines for submitting code & software for further information.

## Data

Policy information about availability of data

All manuscripts must include a data availability statement. This statement should provide the following information, where applicable:

- Accession codes, unique identifiers, or web links for publicly available datasets
- A list of figures that have associated raw data
- A description of any restrictions on data availability

Raw published data for the SciPlex 3 (31), lupus patients (38), glioblastoma patients (44), and statefate dataset (46) are available from the Gene Expression Omnibus under accession codes GSM4150378, GSE96583, GSE148842, and GSE140802, respectively. Data from the cross species dataset (45) is hosted on the BioStudies database of EMBL-EBI under code E-MTAB-6754. A full set of links can be found in their publication. Details on web links for raw 4i melanoma data and all processed datasets can be found in the Data Availability section.

# Field-specific reporting

Please select the one below that is the best fit for your research. If you are not sure, read the appropriate sections before making your selection.

☒ Life sciences ☐ Behavioural & social sciences ☐ Ecological, evolutionary & environmental sciences

For a reference copy of the document with all sections, see nature.com/documents/nr-reporting-summary-flat.pdf

# Life sciences study design

All studies must disclose on these points even when the disclosure is negative.

| | |
|---|---|
| Sample size | We use two cell lines and treated them with 34 drugs and combinations of drugs. Each experiment contains thousands of cells that were profiled with the 4i technology. |
| Data exclusions | Details in Section S4.2 of the Supplement. Specifically, about the exclusion of cells in the subsequent analysis, we state: "Cells tainted by artifacts related to sample preparation and image analysis (e.g., miss-segmentation, detachment during 4i procedure, fluorescent debris) were manually selected using TM's graphical interface and used to train random forest classifiers to systematically exclude cells with similar artifacts from the dataset. Further, cells whose segmentation masks touched image boundaries were also excluded from the dataset." |
| Replication | *Describe the measures taken to verify the reproducibility of the experimental findings. If all attempts at replication were successful, confirm this OR if there are any findings that were not replicated or cannot be reproduced, note this and describe why.* |
| Randomization | We split the data into training and test set for training and evaluating the method. The Online Methods contains a more detailed description of the setup and also how we consider the out of sample and out of distribution settings. |
| Blinding | Blinding was not relevant. |

# Reporting for specific materials, systems and methods

We require information from authors about some types of materials, experimental systems and methods used in many studies. Here, indicate whether each material, system or method listed is relevant to your study. If you are not sure if a list item applies to your research, read the appropriate section before selecting a response.

### Materials & experimental systems

| n/a | Involved in the study |
|---|---|
| ☐ | ☒ Antibodies |
| ☐ | ☒ Eukaryotic cell lines |
| ☒ | ☐ Palaeontology and archaeology |
| ☒ | ☐ Animals and other organisms |
| ☒ | ☐ Human research participants |
| ☒ | ☐ Clinical data |
| ☒ | ☐ Dual use research of concern |

### Methods

| n/a | Involved in the study |
|---|---|
| ☒ | ☐ ChIP-seq |
| ☒ | ☐ Flow cytometry |
| ☒ | ☐ MRI-based neuroimaging |

## Antibodies

| | |
|---|---|
| Antibodies used | A detailed table of antibodies is provided in Section S3 of the Supplements. |
| Validation | *Describe the validation of each primary antibody for the species and application, noting any validation statements on the manufacturer's website, relevant citations, antibody profiles in online databases, or data provided in the manuscript.* |

## Eukaryotic cell lines

Policy information about cell lines

| | |
|---|---|
| Cell line source(s) | We consider two co-cultured primary melanoma cell lines (M130219 and M130429), which were derived from the same melanoma patient from different body sites. M130219 originates from a subcutaneous biopsy taken during treatment with Bimetinib (MEKi), whereas M130429 was derived from a bone autopsy one month after stopping said targeted therapy (29). Described first here: https://doi.org/10.1111/exd.12683 |
| Authentication | The cell lines were gifted from the Mitchell Levesque (University of Zürich/ University Hospital Zürich) senior author of publication that first described the cell lines and not explicitly authenticated. |

| Mycoplasma contamination | Cells were tested for the absence of mycoplasm before use. |
| Commonly misidentified lines (See ICLAC register) | none |

