## [Peer Review File · Nature Methods]

Peer Review Information

Manuscript Title: Learning single-cell perturbation responses using neural optimal transport

Corresponding author name(s): Gunnar Rätsch

Editorial Notes: n/a

Reviewer Comments & Decisions:

Decision Letter, initial version:

Dear Gunnar,

Your Article entitled "Learning Single-Cell Perturbation Responses using Neural Optimal Transport" has now been seen by three reviewers, whose comments are attached. While they find your work of some potential interest, they have raised concerns which in our view are sufficiently important that they preclude publication of the work in Nature Methods.

We will consider looking at a revised manuscript only if further experimental data allow you to address all the major criticisms of the reviewers (unless, of course, something similar has by then been accepted at Nature Methods or appeared elsewhere). This includes submission or publication of a portion of this work somewhere else.

In order to consider the manuscript again, we will require the technical concerns to be addressed, including additional benchmarking. Beyond this, we need you to make a stronger case that this approach is broadly needed for new biological discovery. We hope you understand that until we have read the revised paper in its entirety we cannot promise that it will be sent back for peer-review.

If you are interested in revising this manuscript for submission to Nature Methods in the future, ****please contact me to discuss your appeal**** before making any revisions. Otherwise, we hope that you find the reviewers' comments helpful when preparing your paper for submission elsewhere.

Sincerely,
Rita

Rita Strack, Ph.D.
Senior Editor
Nature Methods

Reviewers' Comments:

Reviewer #1:

Remarks to the Author:

In this manuscript, Bunne et al. present CellOT, a method to predict the transcriptomes of cells exposed to an arbitrary perturbation. They justify their method by asserting the need to predict perturbation responses at the level of individual cells. They furthermore assert that published methods designed to achieve this aim via linear operations in a learned latent space are poorly suited to the problem at hand. Instead, they argue that optimal transport (OT), which has been employed to model cellular differentiation trajectories using time-course scRNA-seq data (Schiebinger et al., Cell 2019), provides a more conceptually apt framework. Issues with model stability and optimization compel them to devise a framework for neural OT based on convex neural networks. They show that the resulting method can predict the effect of a perturbation on cells that were not present in the original training dataset.

The modelling and implementation appear to be well done. However, the conceptual justification for the method is unclear, and it is not immediately apparent what kind of experiment would require a tool like CellOT to interpret the results. These issues are underscored by the nature of the biological insights the authors are able to extract using CellOT in the case studies presented in the manuscript, which appear to be limited. Moreover, comparisons to baseline methods have a number of conceptual and technical issues, and do not convincingly demonstrate the superiority of this new tool over existing methods. Finally, many of the authors' claims about the unique advantages of their method are not supported by the data presented in the manuscript.

Major

1. The authors' central argument is that the destruction of cells in the course of obtaining single-cell measurements prevents us from understanding how individual cells respond to a given perturbation. Yet what is almost always of interest is the aggregate response of cells of a particular cell type or state to a perturbation. It is not clear why a method operating at the resolution of individual cells is needed. The authors assert that "It is crucial [...] to not simply model average perturbation responses of a patient cohort" but what biological question would the paired distribution be needed to answer? The need for a

method operating at single-cell resolution is undermined by the authors' argument that it is necessary to account for subpopulation structure in single-cell data; this can readily be achieved by clustering the data to identify subpopulations at arbitrary granularity and then analyzing the average response within each cluster (for instance, via differential expression).

2. The above issues are reinforced by the observation that most of the biological results presented in the manuscript are not actually obtained from CellOT. For instance, in the extensive case study of the newly acquired 4i dataset (p. 6), most of the biological findings are based on either the abundance of each cell state, or the mean expression of a given marker within each cell state. The authors claim to have "sharpen[ed] the response profiles of the measured drugs," but the meaning of this is unclear: as far as I can tell, the results were not obtained with inferred (predicted) 4i profiles, nor is it clear that it would be desirable to do so. Similarly, it is not clear what has been learned from the analysis of the lupus dataset in Fig. 4. Overall, it is not clear what biological findings CellOT might enable that could not have been made from the raw sequencing or proteomics data itself.

3. The authors evaluate CellOT by comparing it to two baselines, scGen and cAE, on a total of three datasets. There are both technical and conceptual issues in this evaluation.

- o The number of datasets is small, and performance appears to be very variable from one dataset to another. For instance, in Fig. 2, the performance of scGen appears close to random, yet in Fig. 4 it is nearly as good as CellOT. Comparing these methods on a much larger number of datasets would more convincingly establish that there is in fact a bona fide difference in performance.

- o The two primary metrics used to quantify performance, the MMD and L2DS, both seem logical, but it is of note that (to my knowledge) neither has been used in previous work. How would CellOT perform if evaluated on the same datasets as in the scGen paper, using the same metrics?

- o The MMD is calculated on only the top 50 marker genes in the scRNA-seq datasets. This is a tiny fraction of the transcriptome. How sensitive is performance to this threshold (e.g. top-20, top-500, top-2000)? What would performance look like if the MMD were calculated over the entire transcriptome?

- o It would be useful to include the identity and observed baselines in all comparisons (i.e., Fig. 4c).

- o On a conceptual level, I am not sure what kind of biological problem the cross-validation setup employed in the manuscript is a good surrogate for. The model is trained on cells from each state, exposed to each perturbation. This requirement implies that users already have in hand a dataset that would allow them to identify perturbation responses within each cell state. Moreover, it implies that the method would not be able to predict perturbation responses for new cells from a novel state (i.e. leave-cell-state-out), nor would it be able to predict responses for a perturbation that creates a new cell state (cf. Fig. 3c). The cross-validation setup therefore further undermines the justification for CellOT, since it seems the model can only be used to predict perturbation responses that have already been experimentally measured.

4. The authors imply repeatedly that their method can account for variability across samples. In fact, there is no real investigation of how batch effects might compromise the performance of their model anywhere in the manuscript. A unique feature of the Kang et al. dataset is that the control and perturbed samples were all run together in a single sequencing batch, then demultiplexed using demuxlet. This is therefore an artificially weak test of what the authors term the 'out-of-sample' setting, since the cells are in fact from the same sample. This can be seen in the fact that the performance in iid and oos settings is essentially identical (Fig. 4c). Moreover, the dataset is also unrepresentative as a case study, since no prospective application would profile all newly-collected patient samples together with the reference dataset in a single batch. Given that dealing with batch effects is arguably both more difficult and more important than dealing with biological variability across individuals, there is a clear need to evaluate CellOT in a true leave-batch-out setting.

Minor

5. CellOT is applied not to raw scRNA-seq data but to a low-dimensional representation learned by an autoencoder. This raises a number of questions: is dimensionality reduction necessary for the method to work? What would performance look like if CellOT were applied directly to the full transcriptome? Does the choice of dimensionality reduction method matter? The parameters of this autoencoder are tuned by minimizing the reconstruction loss over the full dataset; is this not a form of data leakage between the training and test datasets?

6. The authors tune hyperparameters for CellOT on the evaluation datasets, but not the other two methods. This may introduce a 'continental breakfast included' effect as pointed out by Hu et al., Pac. Symp. Biocomput. 2019. Would performance of the baseline methods improve with hyperparameter tuning?

7. There is no assessment of statistical significance in the evaluation. Can metrics such as the MMD and L2DS be compared statistically, e.g., using the bootstrap?

8. The L2DS is missing for the analysis of the Kang et al. dataset (Fig. 4b).

9. One useful feature that CellOT implements is the ability to score the severity of drug perturbations within each cell state (p. 5) via the mean OT cost. Indeed, this is one of the few biological insights in the manuscript that could not have been obtained without CellOT. However, there are a variety of methods beyond scGen and cAE designed to achieve a similar aim (e.g., Burkhardt et al., Nat. Biotechnol. 2021; Petukhov et al., bioRxiv 2022; Skinnider et al., Nat. Biotechnol. 2021; Chari et al., Sci. Adv. 2021; or simply the distance in latent spaces learned by scGen or cAE). Comparisons to these baselines would be needed to establish that CellOT is uniquely well-suited to this task.

10. Previous methods to predict perturbation responses at the single-cell level implemented a number of other functionalities; for instance, scGen was shown to be among the more accurate methods for batch effect correction (Luecken et al., Nat. Methods 2021). Can CellOT do the same?

Reviewer #2:

Remarks to the Author:

Bunne et al present CellOT, a computational method based on neural optimal transport to model responses due to perturbation with single-cell data.

With the advancement of single-cell technology, this type of study containing multiple perturbed samples is becoming increasingly available and it enables to study response-perturbation relationship with a high resolution. Compared to other methods that often operate on distribution level, CellOT uses cell-level transport map which theoretically would bring in more detailed response modeling that better captures cell-cell variability. CellOT is available as a publicly available Python package. While CellOT is theoretically sound and some validation is included in the current manuscript, more validation is needed especially the ones focusing on cell-cell variability and out-of-distribution generalizations. Comparison to more existing methods should also be added to demonstrate the improvements made in CellOT.

1. For the training/testing split of SciPlex and 4i data, it was explained in the Methods section that “The split is performed on each drug and control condition independently”. So the testing sets in different datasets could have totally different cell type or subtype proportion. Why is the comparison between T (testing control) and testing perturbed when they don’t necessarily represent the same population? In this case, why good overlap between testing sets in the UMAP space (Fig. 2c) is superior?

2. CellOT was only compared to two existing methods cAE and scGEN. There are many more recent methods and more comparison should be added. For example, MELD, PhEMD, and PopAlign.

3. In the learning step, the control and perturbed data are from two different replicates. The feasible set $\{T: T\#(\rho_c) = \rho_k\}$ in the formulation given in Fig. 1c assumes that the mass is conserved. This could be a problem when the cell type ratio in different replicates is different. Though a small discrepancy may be handled by the neural OT, it should be clarified and evaluated in what cases this will become a problem.

4. The main advantage of CellOT is the examination of cell-level variability due to the detailed mapping between cells using optimal transport. The authors have made much effort on demonstrating this on real data. However, the ground truth of responses of individual cells is often not available in real data. It is therefore important to also use simulated data to benchmark the methods.

5. Related to the point above, there are two components of the method: finding the correspondence and learning the perturbation. The validation on the second component is quite extensive. The quality of the OT derived mapping should also be benchmarked.

6. The generalizability (out-of-distribution) performance of CellOT should be further evaluated and clarified. Currently, there is one example and CellOT outperforms the other methods in one case while having slightly inferior performance in another case (Fig. 4e bottom) in the o.o.d. setting. More evaluation of this matter should be added, for example, using the benchmark in Fig. 2 of the scGEN paper. Generalizability is especially of concern here for the OT approach because while OT is expected to find detailed coupling between distributions, it is unclear how well it handles unseen distribution.

7. The software package available at <https://github.com/bunnech/cellot> was tested to work. To improve the usability of the package, I recommend the authors to 1) make it available on PyPI, 2) provide detailed and complete documentation, and 3) setup automatic testing using, for example, pytest.

Minor points

1. In Fig. 2b, why does CellOT achieve lower MMD than the theoretical lower bound? Also, the x-ticks seems to be wrong.

2. In the section "CellOT outperforms state-of-the-art methods in predicting cancer treatment effects", do the top 50 marker genes mean the markers for each cell type? Why not also evaluate on the DE genes between control and perturbed?

3. For "cAE", is it more accurate to use CVAE for conditional variational autoencoder?

4. I tried to run the example numerical experiment on the GitHub page and it took several hours to run. A clarification of scalability and computational cost in terms of time and memory should be added.

Reviewer #3:

Remarks to the Author:

This paper presents an optimal transport-based framework (called CellOT) for learning individual single-cell responses to perturbations. Optimal transport formulation matches probability distributions by learning a coupling/transport map.

The main idea is to learn such a transport map that soft-matches the gene expression profile of control cells with the perturbed cells. This transport map can then produce the distribution of gene expression profiles for that perturbation on a new set of control cells. Directly parameterizing the transport matrix of the primal optimal transport formulation can be a complex optimization problem. However, the paper overcomes this issue by proposing parametrizing the convex potentials (functions f and g) of the dual form of the optimal transport problem. This is achieved by using input convex neural networks (ICNNs), and the transport map is calculated by taking the gradients of function g .

The paper's results demonstrate that CellOT, when applied to predict the responses of cell populations to cancer treatments (using a proteomic dataset consisting of two melanoma cell lines (M130219 and M130429)), outperforms the chosen auto-encoder-based baselines - scGEN and cAE. To quantify the matching distributions of the perturbed cells and ground truth, the paper uses maximum mean discrepancy (MMD) as the evaluation metric (along with l2 distance between drug signatures). CellOT can also learn a transport map from multiple patients that generalizes to new patient samples. Finally, it can model the changes in transcriptome when the perturbations are internal during hematopoiesis.

Overall, the method and application are well motivated, and the presented results are convincing and potentially useful for the community. The neural optimal transport formulation is particularly interesting.

However, the paper's choices and some experimental details could be clarified further.

Major comments:

Choice of evaluation metric?

I briefly checked the scGEN paper, and it seems to be using R2 metric to quantify its results. Another single-cell benchmarking paper [1] comparing simulation methods has proposed metrics like - KS test, mean, variance - for comparing single-cell distributions. Given this prior work, what is the rationale behind choosing the MMD and DS evaluation metrics for the results in this paper?

Choice of the number of features for evaluation?

The paper claims, "Due to the high dimensionality of scRNA data, we report metrics using the top 50 marker genes." Why the top 50 marker genes? scGEN considers up to ~6000 genes for calculating the similarities.

The choice of AE-based baselines - scGEN and cAE seems reasonable. Maybe adding some discussion on non-AE methods like IQCELL [2] could be informative for the reader. Also, some recent single-cell alignment methods using OT (with entropic regularization) have shown reasonable performance. Given that the sample size (number of cells) is reasonable, could they be used for learning a transport map for the proposed task?

Single-cell datasets usually require some normalization before one can perform modeling. Was the normalization/feature selection done separately for training and test splits to prevent information leakage? Or is that not a concern given results on o.o.d and o.o.s datasets?

Finally, I am assuming that CellOT requires hyperparameter tuning to achieve good performance. Were the baseline methods tuned extensively as well? Were hyperparameter grid sizes for tuning baseline models and CellOT the same? Adding some discussion on the robustness of the model to the choice of hyperparameters would be helpful from an application point of view.

Minor comments:

Is the training of the ICNN models reasonably simple? How much time does it take to train a network?

Some typos:

- 1) x-axis in panel b and caption labels of Figure 3 do not match the panel
- 2) ρt in Model subsection 2.1 (should be ρk ?)

References:

- [1] Cao, Yue, Pengyi Yang, and Jean Yee Hwa Yang. "A benchmark study of simulation methods for single-cell RNA sequencing data." *Nature Communications* 12.1 (2021): 1-12.
- [2] Heydari, Tiam, et al. "IQCELL: A platform for predicting the effect of gene perturbations on developmental trajectories using single-cell RNA-seq data." *PLoS Computational Biology* 18.2 (2022): e1009907.

Author Rebuttal to Initial comments

Proposed manuscript revisions for NMEH-A49668, Bunne et al.

Overview reviewers' feedback and intended response/action

The value proposition of CellOT to the field of single-cell biology (Reviewer 1)

In our understanding, Reviewer 1 questions, in general, the value of a computational method capable of correcting/predicting perturbation effects on the level of single cells. Instead, they propose to use methods on aggregated subpopulations or cell types ("this can readily be achieved by clustering the data to identify subpopulations at arbitrary granularity") as this, in their opinion, is sufficient for downstream biological analyses. In particular, they propose to cluster both the control and perturbed populations and then find a pairing scheme to match the clusters detected in the control and perturbed state. Hypothetically, the proposed scheme works in a scenario for which i) the biological heterogeneity in control and treatment is composed of discrete subpopulations ii) the number of biologically meaningful subpopulations in control and treatment, iii) *a priori* knowledge on how to pair control and treatment clusters is available. For the vast majority of experiments, however, these requirements are not fulfilled; perturbation responses, when measured at the single-cell level, often appear as continua and pairing schemes are normally unavailable.

Reviewer 1's clustering approach is close to CellOT in the limit: In the following thought experiment, we intend to show that CellOT is the generally applicable method of Reviewer 1's highly-specialized scheme. Given that single-cell measurements in control and treatment are continuous, we would need ever more clusters to represent cellular phenotypes faithfully. Further, we would need to devise a pairing scheme. Intuitively, we would start pairing clusters based on their feature similarity, i.e., identify pairs of clusters with minimal feature difference, ensuring that overall the difference between the identified pairs is minimal.

As we increase the number of clusters to improve the accuracy of our model, we ultimately arrive at one-cell clusters, i.e., single cells, and are faced with the challenge of identifying the right pairing schemes given all possible single-cell matches. We thus introduce a cost function to match the most similar cells in control and treatment under the constraint of minimizing the total difference between identified pairs. By doing so, we have extended Reviewer 1's method to general applicability and, in fact, have described the workings of CellOT.

CellOT's response prediction is sensitive to cell states: In the following figure, we visualize the learned drug effect for a) CellOT, b) mean corrected, as well as the two baselines c) scGen and d) cAE on 4i cell line data in latent space. The perturbation effect is represented as a vector field (arrows), "pushing" cells of the control distribution (in blue) to the treatment distribution (in orange). We report that the perturbation learned by CellOT accurately shifts control on top of treatment by modulating the drug effect *continuously* as a function of the position in feature space (metrics are reported in Table S2, row 9 of the submitted manuscript). The other three prediction methods are unable to learn the perturbation effects accurately across the whole latent space resulting in mispredicted single-cell data (as seen in Figure below and in Table S2 that measures the accuracy of the predicted distribution).

NB: Reviewer 1's strategy, which is a more sophisticated implementation of "mean corrected", would have identified subpopulations and returned a discrete matching between their before and after state. These clusters could either represent the two cell lines or the responder and non-responders or combinations thereof. Their strategy, however, would have been unable to predict the behavior of those cells in between subpopulations, which display an intermediate drug effect similar to multiple subpopulations.

Proposed manuscript revisions for NMEH-A49668, Bunne et al.

Everolimus response predictions. First two PCs of the protein intensity features.

Improving clarity in the manuscript: It appears that our description of CellOT, the problem it aims to solve and the data it is capable of generating, was lacking given Reviewer 1's feedback. We plan to improve said descriptions in the main and supplementary text. We further plan to include a quiver plot (see Figure above) as a panel in Figure 2, to better highlight subpopulation-specific effects and the value of a single-cell predictions correction over a clustering-based approach.

CellOT-derived results in the manuscript (Reviewer 1)

Further, Reviewer 1 noted that "most of the biological results presented in the manuscript are not actually obtained from CellOT" and that "in the extensive case study of the newly acquired 4i dataset (p. 6), most of the biological findings are based on either the abundance of each cell state or the mean expression of a given marker within each cell state." We find the opposite to be the case. All reported biological findings are based on CellOT-generated data, which in the specific case of Figure 3, were then aggregated to cellular states observed in a population of control cells, for which the drug effects were predicted. *The ability to map drug effects of multiple drug treatments back to the same control cells would not have been possible without CellOT.* Further, the majority of data presented in Figure 3 is single-cell corrected (i.e., "feature vector of a predicted perturbed cell" minus "feature vector of the control cell which was used to predict the perturbed cell"); this requires predictions of the single-cell effects.

Reviewer 1 also states "The authors claim to have 'sharpen[ed] the response profiles of the measured drugs,' but the meaning of this is unclear: as far as I can tell, the results were not obtained with inferred (predicted) 4i profiles, nor is it clear that it would be desirable to do so." We respectfully disagree with this statement. For instance, the UMAP in Figure 3c is wholly based on CellOT-generated data (as are most other panels in Figures 3 and 4). Comparing said UMAP with the

10

Proposed manuscript revisions for NMEH-A49668, Bunne et al.

UMAP generated with mean-corrected single-cell data (Supplementary Figure 6e), we can clearly see that the drug perturbations in 3c are distinct from each other and grouped based on the prevalent mode of action. On the other hand, the perturbations in 6e are heavily overlapping. Thus, we indeed argue that CellOT generates sharpened drug profiles.

Finally, Reviewer 1 asks "[...] what biological question would the paired distribution be needed to answer?" We imagine many cutting-edge biological and translational scenarios which would benefit from paired single-cell distributions, such as the investigation of the molecular mechanism governing the perturbation responses or out-of-ample predictions for patient samples.

We will improve our description and discussion of the results presented in Figure 3 in the main text, as well as mention potential future applications of CellOT in the Discussion.

Out-of-sample predictions and additional datasets (Reviewer 1-3)

We have provided results based on CellOT in an independent-identically-distributed (IID) setting in Figure 2 and 3 as well as in an out-of-distribution (OOD) setting in Figure 4. All three reviewers suggested improvements to our OOD tasks presented in Figure 4, specifically the use of CellOT on alternative datasets to show its robustness as well as its generalizability. We, therefore, plan to add at least one more OOD task based on an additional dataset and use it to further benchmark CellOT against the current state-of-the-art predictive algorithms. We are currently working on preparations.

Gene expression comparison with scGen (Reviewer 1-3)

All reviewers raised concerns about our approach when comparing gene expression predictions between CellOT and the baselines. For high-dimensional datasets, CellOT requires, as do the baselines, some form of low-dimensional embedding. While one can use classical embeddings such as PCA, in this work we follow previous literature and use autoencoders, whose sole purpose is to encode and—with small reconstruction error—decode a cell's feature vector. The evaluation itself is, as mentioned in the manuscript, conducted in the original featurespace (i.e., gene expression space). As the majority of the genes are not affected by each perturbation, and most evaluation metrics rely, to some capacity, on the euclidean metric, we currently evaluate using the top-50 expressed marker genes. Reviewers 2 and 3 both raised concerns about this choice. We plan on including metrics reported on larger feature sets, though the well-known curse of dimensionality demonstrates that such distances lose meaning in higher dimensional spaces. We plan to extend this to reporting metrics on more than 50 genes and add the results to the Supplemental Material.

Inclusion of additional metrics used by the field (Reviewer 1,3)

Reviewers 1 and 3 expressed concern about our choice of metric, maximum mean discrepancy (MMD), used for the prediction accuracy benchmarking of CellOT and the baselines, and proposed a correlation-based metric (R2) as well as population statistics such as mean and standard deviation. In fact, the reported distance between drug signatures (L2DS) is mathematically equivalent to the differences in population means. We will edit the manuscript to make this clear. In preparation for this manuscript, we took special care in identifying the most appropriate (accurate and honest) setup and metric to compare prediction results with actual measured data. As we aim for capturing heterogeneous cell responses, we choose distributional distances which are independent to our model hypothesis, opting to use MMD over the Wasserstein metric, a distributional distance widely used in the field of machine learning. MMD is a strong metric in which low values imply that a pair of distributions share similar values across all moments, i.e., the distributions have the same means, standard deviations, etc. We describe this in the manuscript but will improve the wording to make its implications more obvious.

The selection of setup and the comparative metric in previous work has—in our eyes—severe limitations and does not actually probe the baselines' ability to predict single cells accurately.

Proposed manuscript revisions for NMEH-A49668, Bunne et al.

Correlation-based metric solely captures if a method is capable of predicting cell perturbation responses on the level of a population average. Further, the mean and the standard deviation of a population are a poor approximation of an actual distribution, unless the measured distribution is a neat Gaussian. Single-cell data, particularly if derived from a perturbed system, typically have much more complicated distributions.

Notwithstanding our concerns with the proposed metrics, we will include the standard deviation as well as R2 when benchmarking CellOT with the baselines in Figure 2. We will also improve our explanation on metric choice in the main manuscript. Finally, we will also implement a bootstrapped version of the MMD metric which will enable us to introduce statistical significance testing when benchmarking CellOT to the baselines.

Batch correction (Reviewer 1)

Reviewer 1 would like us to assess CellOT's ability to perform batch corrections (given scGen's ability to do so). CellOT was not designed nor is it intended to be used for batch correction. Instead, we aim to predict the effect of a perturbation. We, therefore, think that the request is outside the scope of this manuscript. We kindly ask you, Rita, to decide whether we should attempt a batch correction task using the suggested dataset from Luecken et al. and whether the results of such an analysis should be included in this manuscript.

Additional baselines (Reviewer 1-3)

Reviewers 1, 2, and 3 requested the implementation of additional baselines (algorithms) to compare CellOT to. They include PopAlign, MELD, phMED, and IQCELL. After careful consideration, we find that they either rely on identifying cell types/clusters (PopAlign), predict which cells will respond to treatment but not how they respond (MELD), are not related to the prediction of perturbation effects (phMED), or are limited to the use of < 6 features (IQCELL). We therefore think that it will be challenging to include additional baseline methods.

We intend to implement PopAlign as an additional baseline and include it in the benchmarking as it is methodologically closest to the task performed by CellOT. At this point, however, it is not clear to us how to modify PopAlign for an OOD task. In any case, we will cite and discuss these algorithms in the main manuscript as related work.

Limitations of OT and CellOT (Reviewer 2)

Reviewers 2 and 3 propose an improved description of the limitations of optimal transport theory in the context of biology and OOD predictions. Further, they would welcome an improved robustness quantification of CellOT (i.e., when does it stop to produce trustworthy results) and "the quality of the OT derived mapping should also be benchmarked."

We therefore will cover the limitations of CellOT and OT better in the Discussion section of the main manuscript. Also, we will generate a dummy dataset and/or introduce noise in one of our own data, with which we will explore and describe the limitations of CellOT prediction ability. A good matching should, in general, yield a small distributional distance to the set of observed cells as well as a small transport cost. For example, a random assignment of untreated cells to treated cells would have a small distributional cost but large transport cost. We will thus add an analysis on the tradeoffs between transport costs and distributional distance for all methods.

Community (Reviewer 2 and 3)

Reviewers 2 and 3 encouraged us to publish CellOT as a software package to the computational single-cell community. We are indeed committed to maximizing accessibility and adoption of CellOT for the whole community. We are actively improving and generating supporting material for CellOT. We will, upon acceptance (if not earlier), populate the current open-source Github repository with

Proposed manuscript revisions for NMEH-A49668, Bunne et al.

documentation, tutorials, and notebooks to not only reproduce our results but also allow users to quickly test their method on a new application.

Hyperparameters (Reviewers 1 and 3)

Reviewers 1 and 3 express concern regarding disadvantageous hyperparameter tuning in favor of CellOT. In fact, quite the opposite is the case. We selected — through an extensive search — the hyperparameters resulting in the most accurate scGen and cAE predictions (details described in Methods). Hyperparameter searches for CellOT, on the other hand, were not required. We have found the method to be quite stable and, in fact, use the same configuration for all experiments. We will extend our explanation regarding hyperparameter selection in the Supplementary to prevent future misunderstandings on the matter, including the important distinction, that unlike the other methods CellOT did not require hyperparameter tuning.

Concerns about train and test splits (Reviewer 2)

Reviewer 2 raises the concern that training and testing splits may not account for individual cell types or that they may be present in different proportions. As common practice, we split the dataset into an 80% train/ 20% test. Statistically, we do not expect major differences as these splits are random. We will provide a quantification report on the similarity of the train and test set for both the SciPlex3 and 4i dataset. The results will be included in the Supplementary of the manuscript.

Varia (Reviewer 1-3)

All reviewers identified amongst others, typos, unconventional abbreviations, and inconsistencies in benchmarking reporting. We thank them for their accurate examination of the manuscript and will rectify the identified mistakes.

Decision Letter, first revision:

Dear Gunnar,

Thank you for letting us know how you would respond to the remaining referee comments for your manuscript "Learning Single-Cell Perturbation Responses using Neural Optimal Transport" (NMETH-A49668B). It has now been seen by the original referees and their comments are below.

Based on the responses you've provided, we'll be happy in principle to publish it in Nature Methods, pending minor revisions to satisfy the referees' final requests and to comply with our editorial and formatting guidelines.

We ask that you include any analyses that were only shown in the previous rebuttal as Supplementary Information and that you clarify and discuss the challenges associated with analyzing the glioblastoma data in particular. Please add clarifications to the text wherever possible to try to reduce the possibility of the concerns raised by the referee being raised by new readers.

TRANSPARENT PEER REVIEW

Nature Methods offers a transparent peer review option for new original research manuscripts submitted from 17th February 2021. We encourage increased transparency in peer review by publishing the reviewer comments, author rebuttal letters and editorial decision letters if the authors agree. Such peer review material is made available as a supplementary peer review file. Please state in the cover letter 'I wish to participate in transparent peer review' if you want to opt in, or 'I do not wish to participate in transparent peer review' if you don't. Failure to state your preference will result in delays in accepting your manuscript for publication.

ORCID

Sincerely,
Rita

Rita Strack, Ph.D.
Senior Editor
Nature Methods

Reviewer #1 (Remarks to the Author):

In their resubmitted manuscript, Bunne et al. have added a number of new analyses that help to clarify the strength and limitations of their method. Some of these analyses are quite convincing. A number of them, however, involve comparisons to artificially weak baselines, while others expose weaknesses in the presented method. In general, the revisions do little to clarify what kinds of biological questions an investigator might use CellOT to answer.

Detailed comments on the points raised in my original review are presented below. A more general observation is that the results rely heavily on examples and visualizations. These include UMAPs, whose deficiencies (Chari et al., doi: 10.1101/2021.08.25.457696) call into question their use in model evaluation, and case studies of individual genes that are well-predicted by CellOT. Quantitative results calculated over entire datasets, when presented, are decidedly more mixed than these examples would seem to suggest. For instance, the r^2 is regrettably not provided for the Kang or hematopoiesis datasets, and shows a mixed picture for the immune and glioblastoma datasets. The r^2 results for the sci-Plex dataset are encouraging, but sci-Plex paper tested 188 small molecule treatments; how were just 9 of these selected, and why are only 5 of those 9 shown in the paper and supplement? What are the results for the other 4?

1. In my original review, I expressed uncertainty about the biological questions that would require a tool like CellOT to answer. The authors' response rests on the premise that single-cell responses to perturbations are heterogeneous, and understanding this heterogeneity is useful to better understand diseases. I think we agree that cell state- or subtype-specific perturbation responses are interesting. I am not convinced that the data presented in this paper supports the contention that predicting paired responses for individual cells is a necessary or desirable approach to understand these responses. This skepticism is augmented by the authors' admission that CellOT (i) only predicts perturbation responses for a few dozen genes among the thousands measured by scRNA-seq, (ii) will only predict perturbation responses similar to those that have already been measured, and (iii) struggles to predict perturbation responses in a realistic example of the exact application the authors suggest (i.e. the glioblastoma dataset).

At a more technical level, the authors' new analyses expose the strengths and weaknesses of CellOT. To my mind, perhaps the most compelling piece of data presented in the resubmission is the inline figure from the rebuttal document (unfortunately not included in the manuscript itself) showing that CellOT dramatically outperforms a very simple baseline at predicting full distributions of gene expression after perturbation, but that this simple baseline does just as well at predicting mean changes in gene expression. I think this experiment, along with the MMD results presented throughout, shows convincingly that CellOT predictions more accurately reflect variability in perturbation responses across individual cells, but it remains very much unclear whether CellOT is better at predicting average responses for a given cell state. After reading the revised manuscript, I am left without a clear sense of the kinds of questions one might be able to answer by predicting variability in gene expression responses to measured perturbations. The authors would certainly need to address this point in order to make this tool meaningful to the community.

2. The authors clarify that the results in Fig. 3 do, in fact, show CellOT-predicted profiles, and argue that these responses could not have been recovered using an average-perturbation baseline. This baseline seems so trivial as to be a straw man: the figure legend (unfortunately no description of the experiment is provided in the Methods) suggests that mean expression in unperturbed cells is averaged over all cells from one of the two cell lines, but the authors acknowledge that the unstimulated population displays subpopulation structure, and it is unlikely any single-cell study would average responses over the entire control dataset regardless of cell states or subtypes. Moreover, the evaluation consists of visual inspection of a UMAP, which as noted above is not quantitative. Broadly, the response does not address the idea that new biological insights can be obtained only through predicted profiles.

3. The authors evaluate CellOT on two new datasets. This is welcome, but does little to assuage concerns about variability in performance from one dataset to another. In the glioblastoma dataset, for example, CellOT is outperformed or equalled by cAE or scGen, depending on the metric. Separately, the

authors' remark that they are running out of single-cell data on which to test their method is difficult to comprehend, given the vast quantities of single-cell data that are publicly available. It is not clear why CellOT would be so specialized as to be applicable only to a tiny fraction of this data.

4. I very much appreciate the authors' efforts to evaluate CellOT on bona fide ood settings, but I am not sure the results establish that handling batch effects is as "out of scope" for CellOT as the authors would wish it to be. For example, in the glioblastoma dataset, the authors find that CellOT is unable to make reliable predictions for a subset of patients, and argue this reflects biological differences between patients, but it strikes me that an equally plausible explanation would be the presence of technical differences between libraries. I will also reiterate that presenting the Kang et al. dataset as an ood evaluation is misleading given that these samples were all in fact sequenced in the same library, and I feel this should at least be noted in the text of the paper, and ideally replaced with a better example.

5. The fact that dimensionality reduction of scRNA-seq data is necessary for CellOT to work, and that the results are reasonably sensitive to the specific choice of embedding, would seem important to at least clarify in the manuscript for potential users.

Reviewer #2 (Remarks to the Author):

The authors have properly addressed my scientific concerns. As for software improvement, the authors mentioned in the rebuttal letter that they are writing documentation and detailed tutorials, plan to make CellOT available on PyPI, and add Python tests to the package. These three tasks should be done before the publication of the paper. I recommend publication of the paper given the above software/documentation improvements are done.

Reviewer #3 (Remarks to the Author):

The revised manuscript and the responses to the reviewers address most of the questions raised during the previous round.

I appreciate that the authors added more datasets and acknowledge the limited availability of datasets for the task.

The authors have addressed all my concerns about the information leakage by re-processing the datasets and fair comparison by incorporating additional metrics and clarifying the choice of hyperparameters.

The choice of parameterized methods and top-k genes has also been clarified.

Author Rebuttal, first revision:

Detailed Response to All Reviewers

Reviewer 1

Bunne et al. have added a number of new analyses that help to clarify the strength and limitations of their method. Some of these analyses are quite convincing. A number of them, however, involve comparisons to artificially weak baselines [...].

In the paper, we benchmark against three published and well-cited methods that represent the current state-of-the-art (scGen, cAE, and PopAlign). In addition, we include two additional baselines allowing us to assess the technical “lower” (Observed) and “upper” bound (Identity). The conducted comparison comprises **all standard metrics of previous work** as well as introduces additional evaluation metrics sensitive to heterogeneity, i.e., MMD. Unfortunately, the reviewer does not provide any evidence to substantiate this claim, and they do not specify why they think that the comparison is based on “artificially weak baselines”.

A more general observation is that the results rely heavily on examples and visualizations. These include UMAPs, whose deficiencies Chari and Pachter (2021) call into question [...], and case studies of individual genes that are well-predicted by CellOT.

We politely but strongly disagree with the reviewer and feel this is a misrepresentation of our work: All of our experiments are backed up with extensive quantitative evaluations using the standard (and non-standard, more sensitive) evaluation metrics. We use examples and visualizations solely to help give the reader insight and intuition into what types of behavior these metrics can capture. Our results thus do *not* “rely heavily” on examples and visualizations.

Each comparison is further supported by a set of quantitative metrics computed on all genes or highly variable genes (computed via ScanPy’s ‘highly_variable_genes’ function) (i.e., l_2 , r^2 , and MMD). In addition, we indeed include “case studies of individual genes” that are intended to help give intuition behind the metrics and in no way act as a substitute for quantitative reasoning. We, therefore, assert that the statement by R1 is misleading.

A detailed list of all instances of UMAP and their contexts within our main analysis can be found below:

- Fig. 2c, f to visualize the mixing of predicted and observed cells. Claims made here are backed up quantitatively by Fig. b, e, and the full set of metrics reported in Fig. S6 and S8. *The UMAPs could be ignored, and all claims would still be valid.*
- Fig. 3b to visualize that we correctly learn pERK response and do so while respecting the distribution of other features. Here we do not interpret the *structure* in the UMAP, rather that the colorings are conserved between whatever structure was learned. These claims could be made under even a random projection (albeit in this setting, it would be harder to interpret). Again, claims about the quality of the predicted cell states are *quantitatively* evaluated in Fig. S6; if the reader understands the reported metrics, this UMAP can be ignored.

- **Fig. 3c** to visualize different perturbation effects (computed from the pairing between unperturbed and perturbed cells recovered by CellOT) of various drugs. The manuscript indeed contains a statement on the distinct cluster structure of the perturbation effects recovered by CellOT (Fig. 3c) vs. the average baseline (Fig. S10d). If necessary, we can support this claim through additional metrics (e.g., quantifications of the neighbor enrichment of perturbation effects recovered through both approaches). In addition, we would like to emphasize that the rest of the analysis in this figure does not depend on any claims made using the UMAP. In particular, the clusters we analyze in the rest of the figure are clustered on control cell states in the original dataspace.

Generally, we share concerns with the reviewer that non-linear projection methods, such as UMAP, can potentially lead to misleading representations of the data. We were careful and deliberate in our use of UMAP as a visualization tool and took care to back up all claims with quantitative reasoning. However, the claims critical of UMAP projections from the manuscript cited in the reviewer's comments by Chari and Pachter (2021) do not appear to have been peer-reviewed and we would like to at least caution the reviewer to exercise special care when using non-peer reviewed work during peer review of other scientific work.

Quantitative results calculated over entire datasets, when presented, are decidedly more mixed than these examples would seem to suggest. For instance, the r^2 is regrettably not provided for the Kang et al. or hematopoiesis datasets and shows a mixed picture for the immune and glioblastoma datasets. The r^2 results for the SciPlex dataset are encouraging.

CellOT outperforms all baselines w.r.t. "quantitative results calculated over entire datasets", i.e., r^2 and l_2 feature means, for four different tasks. This includes two o.o.d. tasks (in particular, the mentioned glioblastoma task). It is unclear to us how these outcomes are interpreted as "mixed" and we believe such claims misrepresent our results.

We are happy to provide any missing r^2 metrics as supplemental material. We would like to note that, during the first round of review, this metric was requested to be included due to legacy reasons. We felt that it is generally an artificially weak metric, evidenced by the fact that it is often maxed out, even by methods that struggle with other metrics (i.e., l_2 and MMD).

SciPlex3 tested 188 small molecule treatments; how were just 9 of these selected, and why are only 5 of those 9 shown in the paper [...]? What are the results for the other 4?

We selected a subset of cancer treatments from the full set of SciPlex3 treatments as showing individual results of 188 treatments seemed excessive. These were selected based on the drug name (preferring cancer drugs) and not on experimental results. If requested, we are ready to provide summary statistics of our considered metrics across all 188 treatments. Additionally, we can provide the results of the missing 4/9 treatments; their omission was due to a mistake in the compilation of the revised document. Both additions could be entered into the supplement. We would like to thank R1 for pointing this out.

1. [...] I think we agree that cell state- or subtype-specific perturbation responses are interesting. I am not convinced that the data presented in this paper supports the contention that predicting paired responses for individual cells is a necessary or desirable approach to understanding these responses.

First, CellOT allows us to model molecular responses in single cells upon perturbation. More specifically, when provided with an unperturbed population of cells (i.e., of an *unseen* sample), it enables us to infer their perturbed states. Using six different datasets, we demonstrate CellOT's ability to capture heterogeneous perturbed states faithfully while outcompeting current state-of-the-art algorithms.

Second, by doing so, CellOT indeed recovers a *pairing* between unperturbed and perturbed states, which is crucial for understanding "cell state- or subtype-specific" molecular mechanisms of perturbations. More specifically, it allows detecting cell state-specific effects of drugs on samples by disentangling subpopulation-specific drug responses (see Fig. 3).

Without such a pairing, one either requires annotation of distinct cell states or subtypes or is restricted to modeling average behavior (whose limitations R1 acknowledges).

Already in the original review as well as this response, R1 hints that perturbations can be modeled simply as average responses within identified cell types, a stance that undermines an active body of research. It is an oversimplification and relies on several strong assumptions, namely that:

- cell states can be appropriately discretized and are not continuous in nature,
- cell types can be accurately mapped across control and treated populations,
- cell types/states do not change under perturbation,
- cell types/states can be identified to a granularity such that responses within them are homogeneous, and
- cell types and their responses generalize across samples.

We believe these assumptions are generally not met but acknowledge that this is subject to current research. In this manuscript, we avoid such simplifying assumptions such as the ones above. CellOT dispenses the need for approximations on population or subpopulation level and does not require additional knowledge on cell types.

The statement further contains false statements and misconceptions, which we address separately below:

This skepticism is augmented by the author's admission that CellOT

i) only predicts perturbation responses for a few dozen genes among the thousands measured by scRNA-seq.

This statement is false. CellOT predicts the perturbation response of single cells across all measured features, e.g., for scRNA-seq all ~1000 highly-variable genes. We can add a specific statement when introducing the method in the first paragraph of the results.

ii) will only predict perturbation responses similar to those that have already been measured.

In general, machine learning approaches are designed to learn patterns and make predictions based on the data they have been trained on. So, if a model has not seen certain types of perturbations before, e.g., a drug with different modes of action, it may struggle to predict their responses accurately. This holds true for CellOT as well as any of the considered baselines.

However, we test CellOT in several o.o.d. settings and demonstrate its ability to capture perturbation responses across different species, cells of varying potency, as well as patients. While out of scope for this paper, it is indeed an interesting direction for future work to combine CellOT with methods that, in theory, could generalize to, for example, unseen drugs. We can include a note on that in the discussion to encourage future directions for the community.

iii) struggles to predict perturbation responses in a realistic example of the exact application the authors suggest (i.e., the glioblastoma dataset).

The reviewer is cherry-picking a single negative example and is ignoring the successful performances reported on three other prediction tasks, comprising two o.o.d. tasks (the cross-species and hematopoiesis dataset) as well as one o.o.s. task (the lupus patient dataset).

The glioblastoma task contains data from seven highly diverse patients and subsequently, both CellOT, as well as the baseline methods, are not able to capture perturbation responses in the o.o.d. setting. Predicting treatment responses to unseen patients (in particular based on a very small cohort) is simply not possible for any current approach that we are aware of. We nevertheless believe including such a “negative” result is important and a good reminder of the challenges ahead of us as a community. Such tasks require additional datasets, larger cohorts, and other technical extensions. While this is an ongoing effort of our team, this is out of the scope of the current submission. We discussed this in the final paragraph of the Discussion section.

1. (continued) [...] Perhaps the most compelling piece of data presented in the resubmission is the inline figure from the rebuttal document [...] showing that CellOT dramatically outperforms a very simple baseline at predicting full distributions of gene expression after perturbation, but that this simple baseline does just as well at predicting mean changes in gene expression.

I think this experiment, along with the MMD results presented throughout, shows convincingly that CellOT predictions more accurately reflect variability in perturbation responses across individual cells, but it remains very much unclear whether CellOT is better at predicting average responses for a given cell state.

We feel this is a misconception by the reviewer. A method with access to the “true” cell type annotation (as the case for baseline modeling the cell type-specific average response) performs *of course* on par with CellOT when considering metrics that can only capture mean effects (r^2 -means and l_2 -means). However, despite its reliance on this additional and manually curated biological information, this “simple baseline” struggles when evaluated with metrics that capture higher moments of the distribution of treated cells (l_2 -std and MMD), i.e., heterogeneity even within both cell lines.

2. [...] Results in Fig. 3 [...] could not have been recovered using an average-perturbation baseline. This baseline seems so trivial as to be a straw man. [...] It is unlikely any single-cell study would average responses over the entire control dataset regardless of cell states or subtypes.

Regrettably, the reviewer appears to ignore the additional analysis provided in the rebuttal that extends the current analysis with a cell type-specific average response computed based on additional cell type annotations (and not “average responses over the entire control dataset regardless of cell states or subtypes”). If helpful and requested, we can add the results and a description of this additional analysis to the supplement.

Again, an accurate identification of cell types is often not possible. During the rebuttal, we thus added the baseline PopAlign, an approach detecting the underlying subpopulation structure in an unsupervised manner (see Fig. 2).

Moreover, the evaluation consists of a visual inspection of a UMAP, which as noted above is not quantitative.

We disagree with this statement as the evaluation is based on other metrics but not the UMAP itself. In addition, please note the discussion above on our general use of UMAPs throughout the manuscript.

3. The authors evaluate CellOT on two new datasets. This is welcome but does little to assuage concerns about variability in performance from one dataset to another. In the glioblastoma dataset, for example, CellOT is outperformed or equaled by cAE or scGen, depending on the metric.

Respectfully, we believe this to be a misleading representation of our results presented in the manuscript.

Across all previous datasets (4i, SciPlex3, hematopoiesis, and lupus patients), CellOT consistently outperforms cAE and scGen in all metrics. This is also the case in the newly added cross-species dataset. CellOT defeats scGen and cAE in all metrics (except it is on par with scGen w.r.t. r^2 feature means, see Fig. 4f). Our analysis further demonstrates that CellOT is capable of predicting bimodal gene activation in an o.o.d. setting, whereas scGen predicts a (biologically meaningless) unimodal expression (Fig. 4g).

As outlined above, the glioblastoma dataset represents a particularly difficult problem for CellOT as well as the baselines. By containing only seven heterogeneous patients, predictions in the o.o.d. settings are challenging or almost impossible.

Separately, the authors remark that they are running out of single-cell data on which to test their method is difficult to comprehend, given the vast quantities of single-cell data that are publicly available. It is not clear why CellOT would be so specialized as to be applicable only to a tiny fraction of this data.

This statement by R1 is misleading. In the rebuttal, we write:

“The preprint of Pleidl et al., (2022) (8) provides a recent overview of standardized datasets, and we have exhausted all o.o.d. listed tasks.”

i.e., we are referring to potential o.o.d. but *not* i.i.d. tasks. The manuscript currently contains six different i.i.d. tasks of various different natures, which should be sufficient to support the claim that CellOT is state-of-the-art in predicting single-cell perturbation responses.

4. [...] In the glioblastoma dataset, the authors find that CellOT is unable to make reliable predictions for a subset of patients and argue this reflects biological differences between patients, but it strikes me that an equally plausible explanation would be the presence of technical differences between libraries.

We thank the reviewer for this comment and agree with it. We will amend the manuscript to include “the presence of technical differences between libraries” as a potential explanation for CellOT’s limited ability to predict a subset of patients.

I will also reiterate that presenting the Kang et al. dataset as an o.o.d. evaluation is misleading given that these samples were all in fact sequenced in the same library, and I feel this should at least be noted in the text of the paper, and ideally replaced with a better example.

We would like to emphasize that throughout the manuscript, we use the term “out-of-sample” when referring to, presenting, or discussing results related to the Kang et al. dataset (e.g., Fig. 4a-c). Further, we specify in the “Result” section how “out-of-sample” and “out-of-distribution” differ (see section “CellOT reconstructs innate immune responses across different species”). Rather than removing our findings derived from the Kang et al. dataset, we believe them to be a valuable addition to the readers of the manuscript as it provides an example of an o.o.s. task.

5. The fact that dimensionality reduction of scRNAseq data is necessary for CellOT to work, and that the results are reasonably sensitive to the specific choice of embedding, would seem important to at least clarify in the manuscript for potential users.

We agree with the reviewer’s comment. CellOT as well as the other baselines require some form of dimensionality reduction for scRNAseq data. We would like to emphasize, however, that the evaluation is conducted on the gene expression profiles, and thus potential sensitivities of embedding choices are captured by the evaluation metrics. We will, however, add a remark on the use of dimensionality reduction to the Discussion of the manuscript.

Reviewer 2

Thank you very much for your positive feedback on our revised manuscript. As you noted, we mentioned in our rebuttal letter that we are writing documentation and detailed tutorials and plan to make CellOT available on PyPI. We are happy to report that we have made significant progress on these tasks and are now in the final stages of completing them.

Reviewer 3

Thank you very much for your positive feedback!

Final Decision Letter:

Dear Gunnar,

I am pleased to inform you that your Article, "Learning single-cell perturbation responses using neural optimal transport", has now been accepted for publication in Nature Methods. Your paper is tentatively scheduled for publication in our August print issue, and will be published online prior to that. The received and accepted dates will be June 28, 2022 and June 23, 2023. This note is intended to let you know what to expect from us over the next month or so, and to let you know where to address any further questions.

Once your paper is typeset, you will receive an email with a link to choose the appropriate publishing options for your paper and our Author Services team will be in touch regarding any additional information that may be required.

Please note that *Nature Methods* is a Transformative Journal (TJ). Authors may publish their research with us through the traditional subscription access route or make their paper immediately open access through payment of an article-processing charge (APC). Authors will not be required to make a final decision about access to their article until it has been accepted. [Find out more about Transformative Journals](https://www.springernature.com/gp/open-research/transformative-journals)

Your paper will now be copyedited to ensure that it conforms to Nature Methods style. Once proofs are generated, they will be sent to you electronically and you will be asked to send a corrected version within 24 hours. It is extremely important that you let us know now whether you will be difficult to contact over the next month. If this is the case, we ask that you send us the contact information (email, phone and fax) of someone who will be able to check the proofs and deal with any last-minute problems.

If, when you receive your proof, you cannot meet the deadline, please inform us at rjsproduction@springernature.com immediately.

Once your manuscript is typeset and you have completed the appropriate grant of rights, you will receive a link to your electronic proof via email with a request to make any corrections within 48 hours. If, when you receive your proof, you cannot meet this deadline, please inform us at rjsproduction@springernature.com immediately.

Once your paper has been scheduled for online publication, the Nature press office will be in touch to confirm the details.

Once your paper has been scheduled for online publication, the Nature press office will be in touch to confirm the details.

Content is published online weekly on Mondays and Thursdays, and the embargo is set at 16:00 London time (GMT)/11:00 am US Eastern time (EST) on the day of publication. If you need to know the exact publication date or when the news embargo will be lifted, please contact our press office after you have submitted your proof corrections. Now is the time to inform your Public Relations or Press Office about your paper, as they might be interested in promoting its publication. This will allow them time to prepare an accurate and satisfactory press release. Include your manuscript tracking number NMETH-A49668C and the name of the journal, which they will need when they contact our office.

About one week before your paper is published online, we shall be distributing a press release to news organizations worldwide, which may include details of your work. We are happy for your institution or funding agency to prepare its own press release, but it must mention the embargo date and Nature Methods. Our Press Office will contact you closer to the time of publication, but if you or your Press Office have any inquiries in the meantime, please contact press@nature.com.

Nature Portfolio journals [encourage authors to share their step-by-step experimental protocols](https://www.nature.com/nature-research/editorial-policies/reporting-standards#protocols) on a protocol sharing platform of their choice. Nature Portfolio 's Protocol Exchange is a free-to-use and open resource for protocols; protocols deposited in Protocol Exchange are citable and can be linked from the published article. More details can found at www.nature.com/protocolexchange/about.

Please note that you and any of your coauthors will be able to order reprints and single copies of the issue containing your article through Nature Portfolio 's reprint website, which is located at <http://www.nature.com/reprints/author-reprints.html>. If there are any questions about reprints please send an email to author-reprints@nature.com and someone will assist you.

Best regards,
Rita

Rita Strack, Ph.D.
Senior Editor
Nature Methods